# Guiding community discussions on human-water challenges by serious gaming in the upper Ewaso Ng'iro river basin, Kenya

Charles Nduhiu Wamucii[1], Pieter R. van Oel[2], Adriaan J. Teuling[1], Arend Ligtenberg[3], John Mwangi Gathenya[4], Gert Jan Hofstede[5,6], Meine van Noordwijk[7,8], Erika N. Speelman[3]

[1]Hydrology and Environmental Hydraulics Group, Wageningen University & Research, 6700 AA Wageningen, The Netherlands.

[2]Water Resources Management Group, Wageningen University & Research, 6700AA Wageningen, The Netherlands.

[3]Laboratory of Geo-information Science and Remote Sensing, Environmental Sciences, Wageningen University & Research, 6708 PB Wageningen, The Netherlands

[4]Soil, Water and Environmental Engineering Department, School of Biosystems and Environmental Engineering, Jomo Kenyatta University of Agriculture & Technology, P.O Box 62000 - 00200 Nairobi, Kenya

[5]Department of Social Sciences, Urban Economics Group Wageningen University, Hollandseweg 1, 6706KN Wageningen

[6]UARM, North-West University, South Africa

[7]Plant Production Systems, Wageningen University and Research, 6700 AK Wageningen, the Netherlands

[8]World Agroforestry (ICRAF), Bogor 16155, Indonesia

*Correspondence to: Charles N. Wamucii (charles.wamucii@wur.nl)*

## Abstract

Water-related conflicts in river catchments occur due to both internal and external pressures that affect catchment water availability. Lack of common understanding of human-water perspectives by catchment stakeholders increases the complexity of human-water issues at the river catchment scale. Among a range of participatory approaches, the development and use of serious games gained prominence as a tool to stimulate discussion and reflection among stakeholders about sustainable resource use and collective action. This study designed and implemented the ENGAGE game (Exploring New Gaming Approach to Guide and Enlighten), that mimics the dynamics observed during the dry season in the upper Ewaso Ng'iro catchment, North West of Mount Kenya. The purpose of this study was to explore the potential role of serious gaming in subsequent steps of strengthening stakeholder engagement (agenda setting, shared understanding, commitment to collective action, and means of implementation) toward addressing complex human-water challenges at the catchment scale. We assessed the type of decisions made during gameplay, the communication dynamics, active participation, and the implication of decisions made on water availability. The results of three game sessions show that the ENGAGE game raised

awareness and provided a recognizable hydro-logic background to conflicts while guiding community discussions toward implementable decisions. The results revealed increasing active participation, knowledge gain, and use of plural pronouns, and decreasing individual interests and conflicts among game participants. This study presents important implications for creating a collective basis for water management and can inform human-water policies and modification of the process behind water allocation rules in a river catchment.

**Keywords:** Participatory approaches, human-water challenges, stakeholder engagement, communication analysis.

## 1. Introduction

Human decisions drive changes in the physical environment, with both desired and undesired consequences for the social-ecological system in space and time. The changes in the physical environment, in turn, influence human behavior, with human adaptation solving or deepening human-environmental crises (Folke et al., 2016; Tilman and Lehman, 2001). Water-related crises experienced by people in watersheds (at local or regional levels), for instance, may be mainly due to competition for water resources between upstream and downstream users, without sufficient coordination. Differences in human perceptions, decisions, and interests between upstream and downstream users drive human-water crises and conflicts (Lesrima et al., 2021; Wiesmann et al., 2000a; Yousef, 2021). Problems, with many interdependent factors that make them very hard to solve, such as the differences in how humans view and interact with the dynamic physical environment, can be described as 'wicked problems' (Arroyave et al., 2021; Defries and Nagendra, 2017; Lawrence et al., 2022; Levin et al., 2012; Rittel and Webber, 1973). Phenomena such as the tragedy of the commons (Dutta and Sundaram, 1993; Ostrom, 1999) are likely to ensue. Addressing the 'wicked problems' of the Anthropocene requires a combination of knowledge and collective action, where both the 'scientific space' e.g. scientists, and the 'non-scientific space' e.g. small-scale farmers interact with earth systems and human societies (Lawrence et al., 2022). For this type of interaction to happen, there is a need to explore and adapt methodologies that strengthen stakeholder engagement toward addressing complex human-environmental challenges. Five interacting phases in the public debate on engaging stakeholders in natural resource management were identified as (a) agenda setting, (b) shared understanding, (c) commitment to goals, (d) means of implementation, and (e) re-evaluation based on monitoring (van Noordwijk, 2019). Participatory approaches have been used in river catchments to bring catchment stakeholders together in an attempt to solve complex human-water challenges (Villamor et al., 2022). A well-known approach to addressing human-water 'wicked' problems is integrated water resources management (IWRM). IWRM is a comprehensive, participatory planning and implementation process for managing and developing water resources in ways that balance the socio-economic and environmental needs of the present and future (Jain and Singh, 2003; Savenije

and Van der Zaag, 2008). Despite the successes in the implementation of IWRM in balancing the social, environmental, and economical issues of a basin or catchment (Obando et al., 2017; Lenton and Muller, 2012; Najjar and Collier, 2011; Scott et al., 2003), some gaps and challenges still exist such as power imbalances, inclusion, lack of common perspectives, collective actions, sustainable collaborations etc (Biswas, 2008; Giordano and Shah, 2014; Godinez-Madrigal et al., 2019; Rahaman and Varis, 2005; Sivapalan et al., 2012; Sokhem et al., 2007).

Firstly, IWRM does not directly account for the dynamics of the interactions and feedback between water and people (Sivapalan et al., 2012). Secondly and most importantly, IWRM typically adopts participatory methodologies such as workshops, focus group discussions, dialogue groups, etc. Such participatory methodologies are limited in the extent to which they promote participants to interact, understand, and digest the human-water 'wicked' problem. This is because the set-up does not promote 'scientific experts' and 'local experts' to directly engage with the 'wicked' problem. For instance, expert workshops might work well where participants have comparable levels of education and common communication styles, but this may not be the case under differences in cultural norms, and power asymmetries that make it more difficult to reach an agreement that satisfies those who do not have power, (Rodela et al., 2019a; Vente et al., 2016). Even stakeholder engagement standards such as AA100AP (Kim et al., 2018) applicable at local level, or (UNSDG, 2022) applicable at national level, among other standards, fail to create a learning space that goes beyond participation and allows stakeholders to directly engage with the 'wicked' problem, testing scenarios in decision-making, and experiential learning for collective action. Bielsa and Cazcarro (2015) underlined the need for innovative ways of participatory approaches for IWRM to achieve its optimal goals.

Given the complexity of human-water 'wicked' problems, there is a need to transcend the scientific space e.g. scientists, modelers, and policymakers, to incorporate the non-scientific space e.g. small-scale farmers, private water suppliers, pastoralists, and traders. This may help in finding and integrating sufficient knowledge, insights on attitudes, and perceptions from various sources to co-create solutions (Norris et al., 2016; Pohl et al., 2017; Worosz, 2022). Serious gaming is an alternative participatory approach and is regarded as a strong transdisciplinary method (Arnab and Clarke, 2017; Cavada and Rogers, 2020; Hobbs et al., 2015; Janssen et al., 2023; Speelman et al., 2021; Speelman et al., 2023). Serious gaming may include amongst others board games, card games, computer games, role-playing games, or a combination of any of these forms (Speelman et al., 2017). The design of a serious game is an iterative process that evolves with the participative process whereby local stakeholders (i.e. 'local experts') are actively involved in defining the 'wicked problem', design of the questions,

simulations, and outputs (Rodela et al., 2019a; Speelman et al., 2014a, 2019a). Compared to the conventional approaches and modelling, where the 'outsiders' (e.g. hydrological modellers and scientists) define the model components depending on the area of interest (Babel et al., 2019; Mayer et al., 2017), the 'outsiders' have no exclusive power to dictate the serious game components. While the conventional models are 'black-boxed' (Kouw, 2016; Melsen, 2022), the gaming process is 'open' and defined in collaboration with stakeholders (scientists and non-scientists) at all stages, from game conceptualization, game refining, to game implementation. This is one of major differences how serious gaming approach differs from other conventional participatory approaches such as workshops. There are different ways to increase engagement of participants during workshops, such as participatory mapping, experimentation with art-based visuals, etc, Basco-Carrera et al. (2017)

In their study, Flood et al. (2018) conducted a review of 43 serious gaming publications and identified the major shortcomings to effective game design and engagement as; one-off engagement (i.e. several game sessions are needed to enhance learning), capturing complexity without overwhelming the stakeholders, future planning (i.e. linking game results to plan an uncertain future). Serious games are also limited on the number of stakeholders who can be involved in a single game session, a constraint that raises the politics of who should attend the game(s) and why? (Edmunds and Wollenberg, 2001; Wesselow and Stoll-Kleemann, 2018). Studies have also reported that social differentiations and power asymmetries have greater influence on the outcomes of a participatory process (Barnaud and Van Paassen, 2013; Mathevet et al., 2014). Both the facilitators and the stakeholders have various degrees in which they can influence the participatory process (Jonsson et al., 2007). Serious gaming can also exacerbate the contests of power due to constraints of simplifying the complex real worlds, balancing the interests of the locals and the 'outsiders', and different perspectives of the present and future (Venot et al., 2022). A co-construction process where the designers and the participants collaborate to define the entire process is seen as a way to improve legitimacy of the participatory process and enhancing multi-stakeholder cooperation (Barnaud and Van Paassen, 2013; Barreteau et al., 2014; Basco-Carrera et al., 2018; Étienne, 2014). In general, the quality of participatory process depends on how biases and interests of all stakeholders, including 'outsiders' are balanced (Biggs et al., 2021; Daniell et al., 2010). As aforementioned, the politics that shape conventional processes (e.g. the influence of the 'outsider') are dealt with in the gaming approach through an iterative process that evolves with participatory modelling (Barreteau et al., 2014; Marini et al., 2018a; Rodela et al., 2019b; Speelman, 2014a; Speelman et al., 2019b). Hence, this study can be viewed to have done something different from the conventional participatory approaches (such as workshops, where 'outsiders' dictate the process) by creating a different type of collaborative engagement and a 'safe environment' for stakeholders.

Due to lack of uniformity in the ways of conducting participatory engagements in IWRM, there is undoubtly a need to explore different collaborative approaches such as serious gaming. The gaming approach can help local stakeholders move beyond individual interests and perspectives to engage in collective action in addressing complex human-environmental issues (Carrera and Mendoza, 2017; Marini et al., 2018b; Medema et al., 2016). Serious gaming can increase active participation, and negotiation among stakeholders, thus potentially leading to collective understanding and actions (Medema et al., 2016; Ouariachi, 2021; Speelman et al., 2014b, 2019c). In the process, researchers/facilitators/data collectors have an opportunity to gather relevant data and observations that can help document the emerging patterns of the human-environmental system under investigation including co-produced solutions to the existing 'wicked' problems. Among the five stages of engaging stakeholders' in natural resource management, the first two (agenda setting and shared understanding) can be readily supported by locally adapted games, but progress has also been reported on the commitment to goals and exploring means of implementation (Janssen et al., 2023).

Improving stakeholder engagement is a prerequisite for any innovative sustainable system of water resource management (Adom and Simatele, 2022; Lim et al., 2022; Loucks and van Beek, 2017). Board games have been reported to stimulate active participation among stakeholders, promote collective understanding, simplify complex issues and systems, and allow stakeholders to directly engage with the 'wicked' problem, and other participants (Bayeck, 2020; Damron, 2019; Jean et al., 2018; Noda et al., 2019; Radzi et al., 2020; Speelman et al., 2014b, 2017, 2019c). Therefore board games can be seen as suitable tools for improving stakeholder engagement in addressing complex human-environmental issues. Communication is one of the social parameters that enable the manifestation of a group strategy, improved efficiency of strategies, and better decision-making (Isaac and Walker, 1988; Orduña Alegría et al., 2020; Ostrom, 2014). In a serious gaming environment, communication during gameplay is a key factor influencing game outcomes (Baijanova, 2022; Neset et al., 2020; Page et al., 2016). Hence, studying communication patterns during gameplay can help evaluate the stakeholders' engagement and interpret emergent game results. Hence, contributing to the body of knowledge on using the serious gaming approach as an 'alternative tool' to addressing complex 'wicked' problems. Studying communication patterns can help study relational logic (value attached on how stakeholders relate to one another) or instrumental aspects (economic perspectives) or both (Githinji et al., 2023). In addition, games can explore multiple levels of internalization of external impacts of individual decisions, based on rules, economic incentives, co-investment, peer pressure to reduce one's footprint, or genuine concerns for impacts on others (van Noordwijk et al., 2023).

Games can pose a challenge to the players who remain selfish as long as they only consider their direct interests, but emergent collective action can bring new solutions.

The purpose of this study was to assess the potential role of the ENGAGE game in strengthening stakeholder engagement toward addressing complex human-water challenges of a river catchment. Using a board game, gaming sessions were organized involving various stakeholders from upstream, midstream, and downstream zones of a river catchment. Our case study was the Upper Ewaso Ng'iro river basin in Kenya, a catchment that experiences complex human-water challenges leading to annual conflicts between upstream and downstream societies (Gichuki, 2006; Kiteme, 2020; Lanari et al., 2018; Liniger et al., 2005; Mutiga et al., 2010; Wamucii et al., 2023). The key research question to be answered in this study was: To what extent does a gaming approach strengthen stakeholder engagement in, and shared understanding of, the human-water challenges as presented in the board game? The ENGAGE board game was used to model or mimic the 'real life challenges' experienced in the case study site.

## 2. Methodology

To assess the extent of stakeholder engagement during gameplay and whether the ENGAGE game guided decisions towards addressing human-water challenges presented in the board game, the following game variables were pre-identified as key research items to be investigated: (i) the type of decisions made during gameplay, (ii) the type and direction of sentiments as players made various decisions, (iii) active participation among players, and (iv) and implications of decisions made on water availability of the board game system - explored in the solution space defined by all possible responses to the rules of the game (Speelman, 2014b), and developed by carrying out a large set of simulated runs of the game. The solution space of the board game elements was developed to determine the realm of possibilities of participant choices in the ENGAGE game. The possible ranges (the minimum and maximum limits of game results) were explored in the modeled solution space. The overall performance of the game was assessed by plotting the actual game results within the solution space.. The communication analysis focused on the subtractive dynamics (i.e. sentiments revealing tension, conflicts, and selfishness), versus constructive dynamics (i.e. sentiments revealing cooperation, positive collaboration, knowledge gain, and collectiveness).

We hypothesized that engaging participants in a serious game that mimics 'real-life challenges' on complex human-water dynamics would trigger debates on possible alternatives to the human-water challenges presented in the board game. This assumption borrows from the literature that opines that 'problems' or dilemmas activate 'counterfactual thinking' (i.e. thoughts about alternatives to the problems) which can be directly linked to

behavioral changes (Epstude and Roese, 2008). Therefore, we recognize that subtractive dynamics are important triggers of stakeholders' reactions, which can influence decision-making during gameplay. Using this logic, we argue that communication patterns can reveal subtractive dynamics during gameplay in the form of tension, conflicts, and selfishness. As a result, it was important to evaluate how subtractive dynamics triggered the behavior of players during gameplay (based on decisions made). This was assessed based on the extent to which players were collaborating, cooperating with set rules during gameplay, use of the plural ('we' rather than 'I') pronouns, and knowledge gain. Nevertheless, the subtractive dynamics were expected to decrease with the build-up of constructive dynamics during gameplay. Three game sessions representing three different sub-catchments were used to explore emerging patterns during gameplay. The game sessions were video recorded to allow post-game analysis of sentiments.

### 2.1. Case study area

The case study area is the upper Ewaso Ng'iro river basin - North-West of Mt Kenya forested water tower (Fig 1). It is located 180 km north of Nairobi city, between (0.14°N to -0.09°S Latitude and 37.03°E to 37.28°E Longitude). It has a climatic gradient with precipitation ranging from 1500 mm yr$^{-1}$ in the humid upstream zone to 350 mm yr$^{-1}$ in the arid downstream zone (Mungai et al., 2004). Population densities range from 800 persons km$^{-2}$ in the upstream zone to less than 20 persons km$^{-2}$ in the downstream zone.

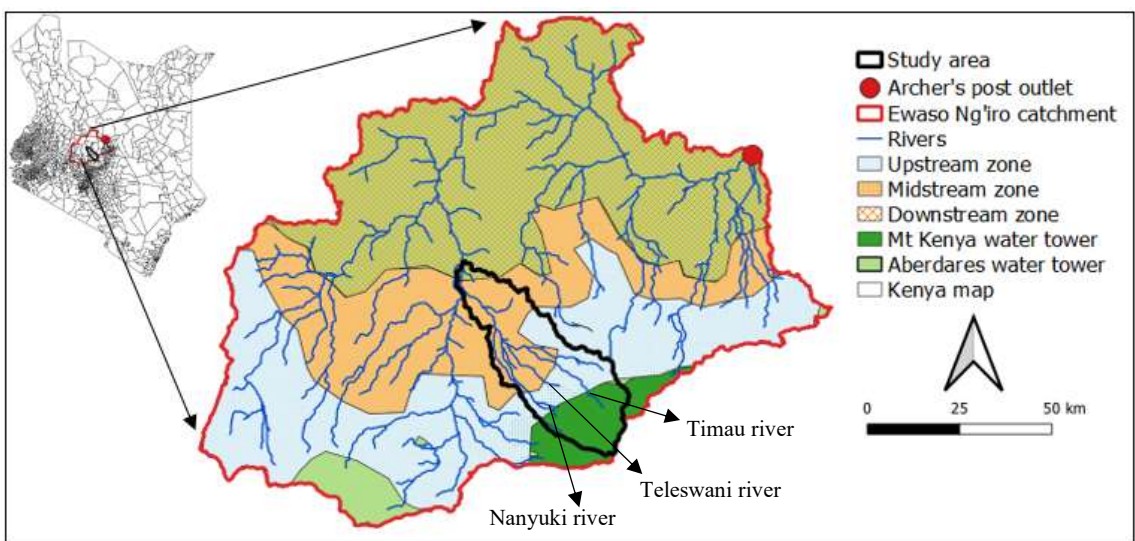

**Figure 1.** Case study area.

In the upstream and midstream zones, small-scale and large-scale agriculture is mainly practiced while in the downstream zone, pastoralism and wildlife-oriented tourism are key activities. The rivers in the sub-basins are managed by the community-based water resources user's association (WRUAs). WRUAs are the link between

water resources and the livelihoods of local communities in a river basin (Richards and Syallow, 2018). WRUAs are legally recognized as community-based associations for the collaborative management of water resources and the resolution of conflicts concerning the use of water resources at the local level (The Water Act, 2016). This study focused on three river sub-basins; Nanyuki river - 95 km, Teleswani river - 30km, and Timau river – 45 km (Fig 1).

The water availability in the catchment is influenced by changes in land-use and climate (Wamucii et al., 2021). The aridity in the catchment changes drastically between the upstream (humid) zone and downstream (semi-arid/arid) zone within a short distance of 40 to 50 Km (i.e. distance from the forested water tower to dry downstream areas) (McCord et al., 2015; Ngigi et al., 2007). The changes in the downstream water availability are attributed to the river water abstractions in the upstream zone (e.g. water used in irrigation, domestic and urban water supplies) (Laikipia Forum, 2021; Gichuki, 2006; MKEWP, 2017; Orendo, 2000; Wamucii et al., 2023). Hence, reducing the downstream hydrological flows, and conflicts emerge when downstream communities cannot adapt to the changing hydrological conditions (Kiteme, 2020; Liniger et al., 2005; Wiesmann et al., 2000a).

The major issues in the catchment are therefore identified in this study as; reduced dry season flows, violent water-related conflicts that intensify during dry seasons, increasing water demand due to human population growth, and agricultural land expansion, among other issues (Kiteme, 2020; Mutiga et al., 2010, 2011; Ngigi et al., 2007; Wamucii et al., 2023; Wiesmann et al., 2000a). Violent conflicts exist between different water users at different levels: upstream versus downstream water users; competing irrigators; agro-pastoralists versus pastoralists; users versus authorities, environmentalists etc (Aarts, 2012a; Ehrensperger and Kiteme, 2005). Wiesmann et al. (2000) noted that the upstream communities lack awareness of the magnitude of downstream effects caused by their activities. This indicates varying perspectives between upstream and downstream communities on human-water issues, hence complicating the management of water resources.

For over 30 years, the WRUAs have received support and capacity building to improve water resources management and governance from both government institutions and non-governmental organizations. This type of support is mainly done through the common approaches involving: workshops, stakeholder discussions, focus group discussions, etc. WRUAs face various challenges including weak enforcement of policies/laws, water abstraction regulations, water metering requirements, protection of riparian corridors/forested areas, etc. These challenges can relate to a lack of collective action due to the individualistic nature of the communities they represent. In addition to the lack of collective actions by the communities, studies have shown that climate change, rapid changes in land-use systems, and societal changes such as population increase, constantly challenge the

ability of WRUAs to modify rules for water allocation (Aarts, 2012b; Dell'Angelo et al., 2014; Lesrima, 2019). With increasing violent water-related community conflicts, the national government reacts by closing water intakes (both legal and illegal) in the upstream zone in desperate attempts to resolve the downstream-upstream conflicts. This temporarily acts as a solution to downstream river flows but negatively affects the livelihoods in the upstream zone due to the termination of water for irrigation. In addition, such national government decisions affect other key amenities such as health facilities, schools, and several industries that are already connected to both legal and illegal water intakes. Given the context above and to explore possible alternatives to WRUA management styles (especially increasing stakeholder engagement), developing and testing an alternative participatory approach such as a serious game was considered timely.

### 2.2. Game conceptualization

This stage involved gathering all possible ideas, to help in drafting a serious game that mimics the context of the case study area. We conceptualized a board plus role-play game, that mimics the complex human-water challenges experienced in the Upper Ewaso Ng'iro catchment, especially with an understanding of how the human-water system works (Wamucii et al., 2023). Relevant catchment issues were sought from publications highlighting the major causes of the changing hydrological conditions and the water conflicts in the upper Ewaso Ng'iro catchment. This was further reinforced by identifying the actors, resources used by actors, key dynamics, and interactions in the case study site, in an approach that is commonly referred to as the ARDI approach (Actors, Resources, Dynamics, Interactions) that directly engages stakeholders in the design and development of the serious game (Etienne et al., 2011). Focus group discussions were also carried out with the communities in the three sub-catchments. The community discussions were helpful in widely discussing the ideas and components included in the board game. The selection of participants and mobilization was done through respective WRUAs in the three sub-catchments. The ENGAGE game as developed and implemented in this study is summarized below and explained in detail in Supplement 1.

### 2.2.1. Description of Boardgame and players

ENGAGE (i.e. "Exploring New Gaming Approach to Guide and Enlighten") is a type of Board plus role-play game (see Supplement 2) that seeks to increase collaborative decision-making in the river basin through experiential learning. The goal of the game is to engage and stimulate discussions and learning among participants. There are a total of ten active game participants per game session:

- 2 participants representing the upstream agricultural community
- 4 participants representing the midstream agricultural community
- 2 participants representing the pastoralists in the downstream zone
- 1 participant plays the role of implementing local water regulations (i.e. WRUA)
- 1 participant plays the role of the national government (imposing rules and fines).

The declared individual goal for the eight land-user participants is to win a game round by accumulating the largest sum of money (profits) at minimal water-related conflicts.

### 2.2.2. Game mechanics

The ENGAGE game mimics the dynamics observed during the dry seasons in the upper Ewaso Ng'iro catchment. The river network (i.e. marbles on boardgame) connects the communities as water flows from the forested Mt Kenya water to the downstream areas. There were two phases in the implementation of the ENGAGE game in this study. Phase one mimics reality, whereby individual values and preferences of the players were allowed to shape the game results. The first two or three rounds were considered sufficient for players to learn from individual decisions and consequences. In the second phase (i.e. a final round or 'reflection' round), the players were guided to reflect on the game results and experiences in phase one and think objectively about what could be the potential solutions to the human-water challenges observed in phase one. There are no maximum rounds of the ENGAGE game, players can continue playing as long as they are willing. However, in this study, four rounds were considered sufficient given the time factor which averaged 2.5 hours per game session in each sub-catchment (i.e. after four rounds). The ENGAGE game as implemented in this study was relatively closed and strictly followed the rules set out in Supplement 1. The rules remained relatively the same in all game rounds apart from the agricultural lands expansion that evolved in the succeeding game rounds. The external observers were also allowed in the game sessions and included persons not directly involved in the playing of the game, but were instrumental during debriefing sessions. More information about the conceptualization and application of the ENGAGE game is given in Supplement 1.

### 2.2.3. Key actions and key outcomes expected in the game


The upstream communities earn their livelihoods from arable agricultural activities; hence they will start by clearing natural vegetation to create cropland areas. Supplemental irrigation is a key decision for the opened agricultural patches. For every two patches cleared for agricultural activities, one marble will be permanently lost from the river network. Agricultural households have the choice of investing in water storage or directly abstracting


available water from the river network. In the downstream zone, the pastoralist households are concerned with the availability of water and grazing area for their livestock. With declining water resources, pastoralists must make quick decisions including selling their livestock or migrating in search of water. One decision is to migrate preferably upwards (as there is the presumption of both sufficient pasture and water in the upslopes).

### 2.2.4. Potential impact on water resources availability


In the first round, participants play the game under an assumed 'normal' climate scenario (i.e. with a maximum of 100 marbles). In the subsequent rounds, a dice is used to determine the exogenous conditions and hence the number of marbles to be placed on the board game (i.e. ranging between 70 and 100 marbles). The water is required for crop irrigation (i.e. in the upstream and midstream zones), household consumption (in all three zones), and livestock production (i.e. in the downstream). However, as marbles get abstracted, the length of the river network


reduces and hence the river starts drying up from the downstream zone upwards.

### 2.2.5. Possible reactions expected by actors and feedback

The agricultural activities intensify in the upstream zone affecting the water balance due to increasing demand for direct water abstractions. The game participants may react by investing in rainwater harvesting or collectively agreeing on water rationing during gameplay. The effects of changes in the water balance are however most heavily


felt in the downstream zone. With time the river dries up from the downstream zone upwards. This forces the downstream community to go upstream to find out where the water has gone. This causes massive destruction of crop fields as pastoralists migrate with their livestock, fuelling intensive conflicts. The authorities react by destroying all water intakes and imposing heavy fines on illegal water users.

### 2.3. Game pre-testing and validation


Game validation comprises a process of building arguments to support (or challenge) the claims, content, and outputs of a game (Hummel et al., 2017). Involving and learning from key stakeholders in the validation is a common practice to reinforce trust, and ownership and to address the external and internal issues of a serious game (Jackson, 2012; Redpath et al., 2018). Three steps were followed in the validation of the game in this study (Fig 2). The first step involved the conceptualization of a serious game as described above. This was followed by pre-

testing the draft game with fellow game designers working on forest-water-people issues elsewhere in the tropics (Van Noordwijk et al. 2020). The pre-testing allowed for assessing the playability of the game and suitability of the game in answering research questions. The final step involved conducting field trials with the communities in the Upper Ewaso Ng'iro river catchment to validate and adapt the final version of the serious game.

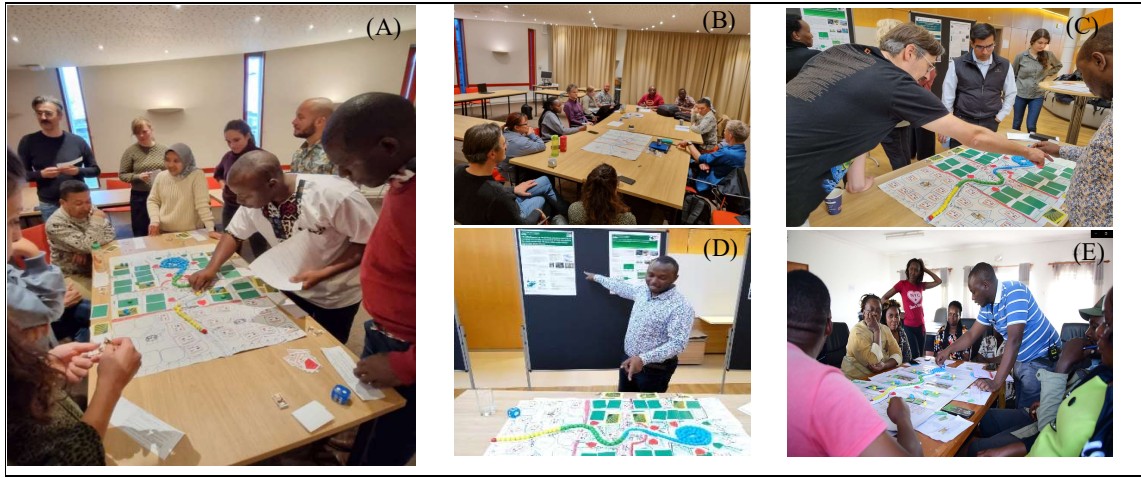

**Figure 2.** Pre-testing the game with Ph.D. students in Wageningen university (A&B), presentation of the game
during the SESAM project outreach day on April 12th, 2022 (C&D), and one of the validation sessions with the targeted communities (E).

### 2.4. Data collection

#### 2.4.1.    Documenting gameplay decisions

A data collection sheet was used to document key decisions made during game sessions such as the number of agricultural patches opened, the amount of water extracted from the river system, the amount of water harvested if any, profit made at the end of each game round, number of livestock at the start and end of each game round. Other data collected during game sessions included: the impact of climate variability on water availability in the board game (i.e. climate variability was mimicked by throwing a dice at the start of a game round that determined
the flow available for each round), water demands in the households and urban towns, and net water availability at the end of each game round.

#### 2.4.2.    Analyzing communication patterns

As participants engaged with one another and made various decisions, their verbalized sentiments were documented to evaluate the emergent patterns in the communication which was used to evaluate stakeholders'
engagement during gameplay. For each of the sentiments extracted, the following issues were considered:

    i)        The direction of each sentiment (i.e. whether the sentiment was directed to the facilitator, to other participants, or as a 'spontaneous' reaction from the board game outcomes).

i)        The nature of the sentiment (i.e. subtractive and constructive dynamics). This involved scoring each sentiment against the game dynamics provided in Table 1.

The subtractive characteristics were scored with values ranging from -5 and 0 and the constructive characteristics were scored with values ranging from 0 and 5 (see Table 1). The summation of both subtractive and constructive dynamics provided the overall status of each game round. The maximum subtractive dynamics had a total of -15 points, and the maximum constructive dynamics had a total of 20 points (Table 1). Each game round resulted in a variety of sentiments and their patterns were explored within this pre-defined scoring range (i.e. -5 and +5). This 

was important to evaluate the communication patterns in the different game rounds and what that means in relation to stakeholders' engagement. The subtractive and constructive dynamics (Table 1) were based on the observation manual for collective serious games (Daré et al., 2021). The extraction of sentiments was done manually through post-game video analysis.

At the end of each game sessions, post-game feedback sessions were also conducted where participants were 

allowed to give their feedback and key lessons on the game sessions. This qualitative feedback was useful in understanding participants perceptions and reflections, which were critical in qualitatively discussing the game results of this study.

**Table 1.** The elements used to monitor game dynamics. The scoring was done by the first author who was the main facilitator of all game sessions.

| Game dynamics | | Sentiment scoring details | Minimum and maximum scores | Total scores |
|---|---|---|---|---|
| **Substractive dynamics** | **1. Tension** | The extent to which the sentiment indicates the participant is worried, or uncomfortable with the game dynamics during the game session. The sentiment can express the tension e.g. one participant is uncomfortable/worried with the action of the other but no visible conflict is observed yet. *(Score 0) - No tension was detected at all. (Score -1) - Very low-level tension. (Score -2) - Slightly tensed. (Score -3) - Somewhat tensed. (Score -4) - Moderately tensed. (Score -5) - Extremely tensed.* | Minimum score = -5 Maximum score = 0 | Substractive dyanmics minimum score = (-)15, and maximum score = 0 |
| | **2. Conflict** | The condition in which the sentiment is made in regard to the level of aggression of participants and/or visible conflicts during the game session. A visible conflict can be observed during gameplay, e.g., denying access to a resource resulting in the deaths of livestock, losses, etc. Note that when there exists a visible conflict, the scoring for tension is set at a minimum score of (-)5. *(Score 0) - No aggression/visible conflicts were observed at all. (Score -1) - Very low aggression/visible conflicts. (Score -2) - Slightly aggressive. (Score -3) - Somewhat aggressive. (Score -4) - Moderately aggressive. (Score -5) - Extremely aggressive/visible conflicts were observed.* | Minimum score = -5 Maximum score = 0 | |
| | **3. Selfishness** | The extent to which the sentiment indicates the level of selfishness/ self-seeking behavior during the game session, e.g. when a player refuses to share a resource. *(Score 0) - No selfishness was detected at all. (Score -1) - Very low selfishness. (Score -2) - Slight selfishness. (Score -3) - Some selfishness. (Score -4) - Moderate selfishness. (Score -5) - Extreme selfishness.* | Minimum score = -5 Maximum score = 0 | |
| **Constructive dynamics** | **1. Cooperation** | The extent to which the sentiment indicates the level of positive cooperation with the existing rules set by other participants or facilitators during gameplay. *(Score 0) - No cooperation at all. (Score 1) - Very low cooperation. (Score 2) - Slight cooperation. (Score 3) - Some cooperation. (Score 4) - Moderate cooperation. (Score 5) - Extreme cooperation.* | Minimum score = 0 Maximum score = 5 | Constructive dynamics minimum score = 0, and maximum score = 20 |
| | **2. Collaboration** | The extent to which the sentiment indicates the level of positive collaboration towards solving the issues at hand during gameplay. For instance, two participants or more agree to reduce water abstraction in a given round to allow the river to flow to the downstream zone. *(Score 0) - No collaboration at all. (Score 1) - Very low collaboration. (Score 2) - Slight collaboration. (Score 3) - Some collaboration. (Score 4) - Moderate collaboration. (Score 5) - Extreme collaboration.* | Minimum score = 0 Maximum score = 5 | |
| | **3. Knowledge gain** | The extent to which the sentiment indicates a new realization or knowledge gained by the participants during gameplay. *(Score 0) - No knowledge gain was detected at all. (Score 1) - Very low knowledge gain. (Score 2) - Slight knowledge gain. (Score 3) - Some knowledge gain. (Score 4) - Moderate knowledge gain. (Score 5) - Extreme knowledge gain.* | Minimum score = 0 Maximum score = 5 | |
| | **4. Presence of plural pronouns** | The presence or absence of plural pronouns in the sentiment. Examples of plural pronouns included: 'we', 'us' 'our' 'their' etc. Note that the study focussed on inclusive 'we' as described in (van Noordwijk et al. 2023). *(Score 0) - Absence of plural pronouns. (Score 5) - Presence of Plural pronouns. Note that a score of 5 (in cases of the presence of plural pronouns) was preferred to ensure the fitting of all the constructive dynamics on a similar scale of 0 to 5.* | Minimum score = 0 Maximum score = 5 | |

**2.4.3.** Modeling the game solution space

A system dynamic model of the board game elements was developed to determine the solution space of all possible participant choices in the ENGAGE game. This was important to establish the envelope within which ENGAGE games operate, by understanding the minimum and maximum values of the various game metrics. Figure 3 illustrates the system dynamics modeling of the board game elements. The possible ranges of game outcomes were explored in the modeled solution space. The solution space was constructed by a total of 1000 runs (more details are provided in Supplement 3).

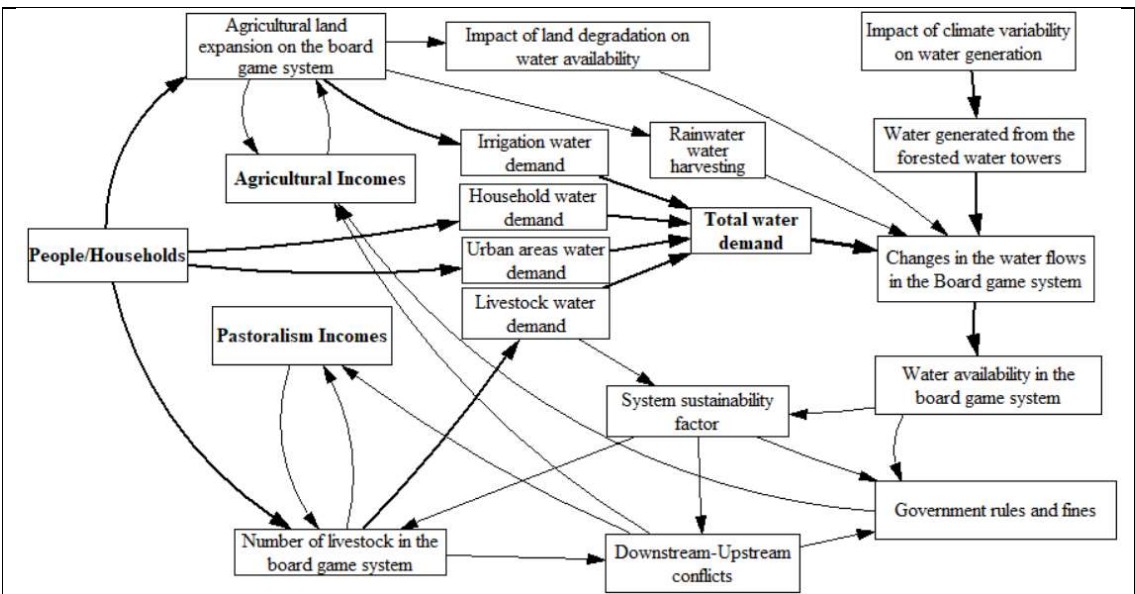

**Figure 3.** The schematic representation of the system dynamics modeling of the elements of the board game

Water resources in the board game are generated from the Mt. Kenya forested water tower and a river network is represented by 100 marbles (string-connected) as illustrated in Supplement 2. The 100 marbles mimic the 'normal' climatic scenarios, hence 100% of water availability. To represent interannual rainfall variability, a dice was thrown and the value obtained determined the flow for the game round indicated by the number of marbles between 70 and 100 (Fig S1). Within the board game, there is competition for water due to various water demands such as water for irrigation, household consumption, livestock, and urban water demand. Water availability was accounted as the difference between the water generated from the water tower and total water demand. Rainwater harvesting was considered as 'additional water' for the board game, as this was done during the transition of game rounds. A detailed description of the development of the solution space is provided in Supplement 3 and Table S1.Results

**2.5.** **Decisions made during game sessions**

For the agricultural community in the upstream and midstream zones (i.e. players 1 to 6), the results from the three game sessions showed that the players adopted a systematic approach to opening up agricultural patches. The

players began by opening up only a few agricultural patches in the initial rounds, but this increased in the succeeding rounds as shown in (Fig 4A, 4B, and 4C). River water abstractions increased with increasing numbers of agricultural patches, especially in the midstream zone (i.e. players 3, 4, 5, and 6) - a relatively dry zone (Fig 4D, 4E, and 4F). Water harvesting was increasingly selected in the succeeding game rounds (Fig 4G, 4H, and 4I). In the downstream zone, the game results revealed unsystematic stocking of livestock units by pastoralists (i.e. players 7 and 8). The number of livestock that survived within the board game system (at the end of each game round) was observed to be equal to or fewer than the available stock at the start of the game. Toward the end of a game session, we observed stability in the number of livestock that survived in the system (Fig 4J, 4K, and 4L). This stability coincided with rainwater harvesting adopted by all the players in the succeeding rounds. The key characteristics of the final round included; moderate water availability in the board game, reduced upward migration of livestock units, reduced losses of crop damages, minimal or no conflicts, reduced government interference and fines, etc.

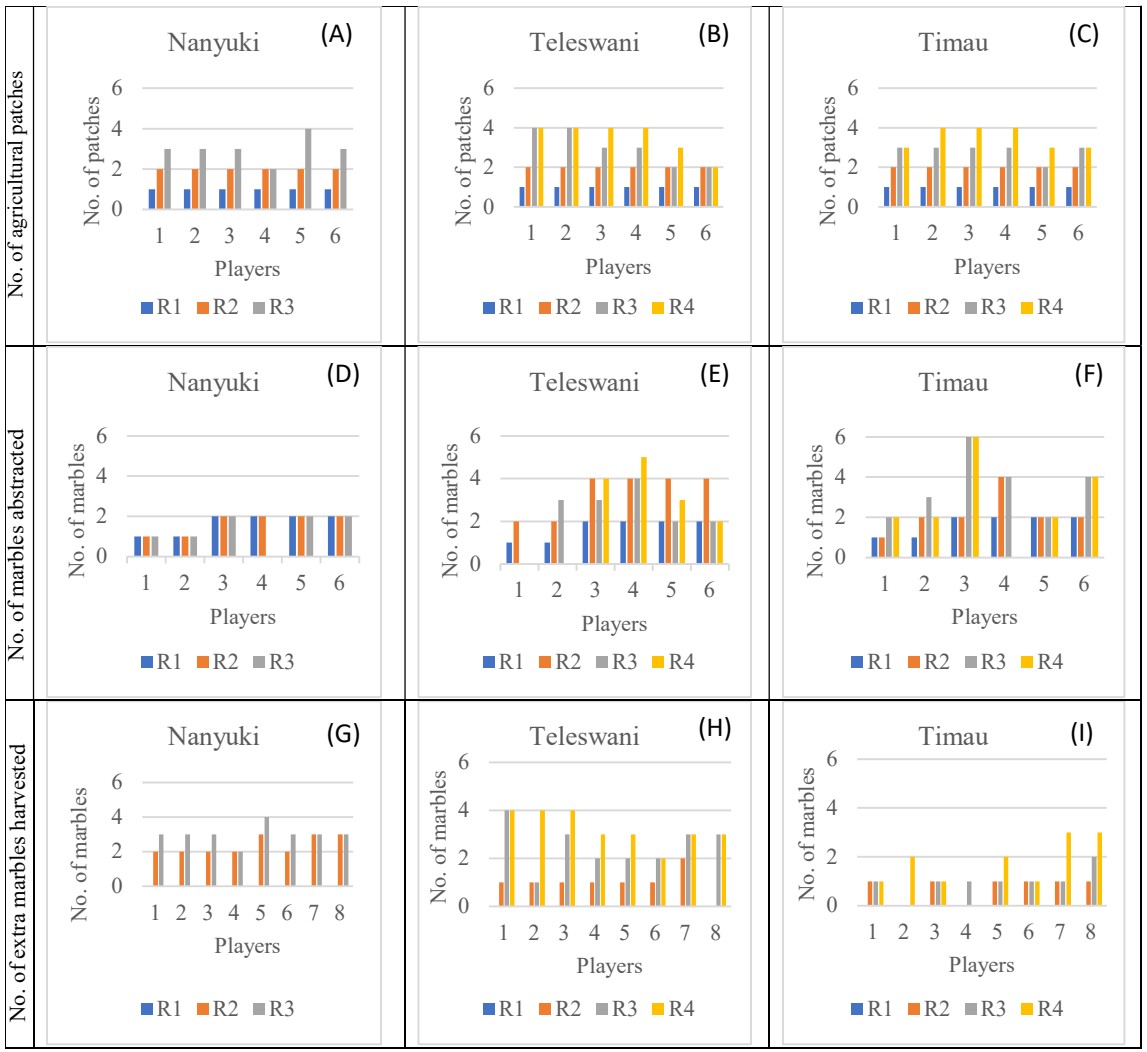

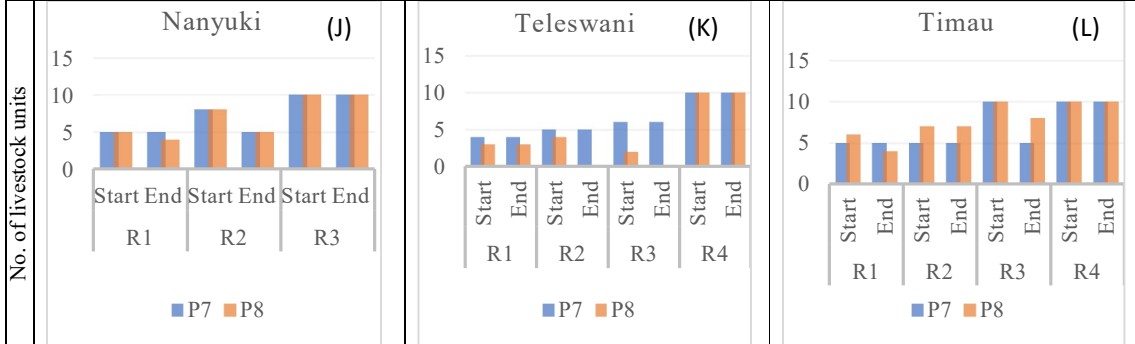

**Figure 4.** Summary of game results in the three sub-catchments. Number of agricultural patches opened per player (A, B, and C), Number of marbles extracted from the board game per player (D, E, and F), Number of extra marbles available per player (harvested during rainy season) at the start of each round (G, H, and I), Number of livestock at the start of the game and the end of the game (J, K, and L). 'R' represents game rounds.

The profits earned seem to have an increasing trend among the agriculturalist players (i.e. players 1 to 6) in the succeeding rounds, which coincided with the land expansion on the board game (Fig 5). On the contrary, profits earned by pastoralists (i.e. players 7 and 8) varied and it was dependent on the number of livestock units that survived in the board game system for each round. There were some game rounds where pastoralists did not sell livestock units, resulting in debts or negative net profits (Fig 5A and Fig 5B). Interestingly, the rate of change in profits between the game rounds was lowest among pastoralist players compared to the agriculturalist players (Fig 5). In the three-game sessions, the average change in profits ranged between 0% and 65% for pastoralists, while for agriculturalists, the average change in profits ranged between 48% and 128%. A general observation is that higher profits were realized towards the final rounds when the boardgame system was relatively stable. Profits were observed to be affected by several factors during gameplay including the number of agricultural patches (and whether irrigated or non-irrigated), the number of livestock units in the board game, government fines, crop losses due to pastoralist migration, corruption, players' debt during gameplay, etc.

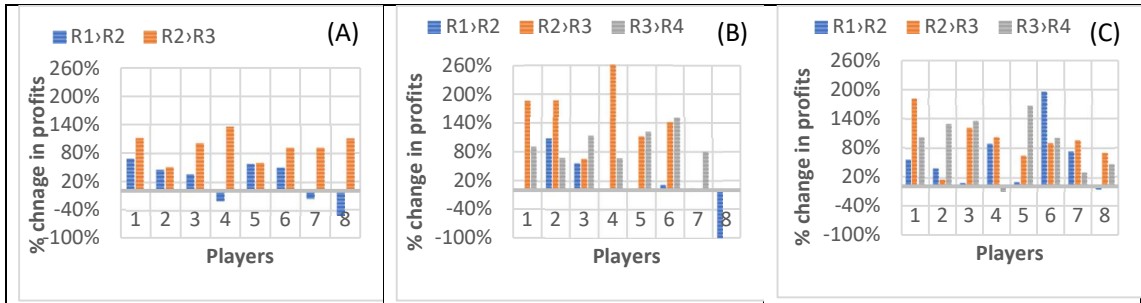

**Figure 5.** Percentage in profits between consecutive rounds in the different game sessions. (A) Nanyuki, (B) Teleswani, and (C) Timau.

### 2.6. Communication analysis

#### 2.6.1. Participation during gameplay

During game sessions, participants engaged one another as they interacted with the dynamics of the board game. A total of 181 sentiments were extracted from the video records of the three-game sessions; 44 sentiments from three rounds in Nanyuki, 83 sentiments from four rounds in Teleswani sub-catchment, and 54 sentiments from four rounds in Timau sub-catchment. The results of this study show that most of the sentiments raised in a game round were mainly directed to other participants (Fig 6A, 6B, and 6C). This indicates that the gaming approach stimulated and sustained active participation among the participants throughout the game rounds.

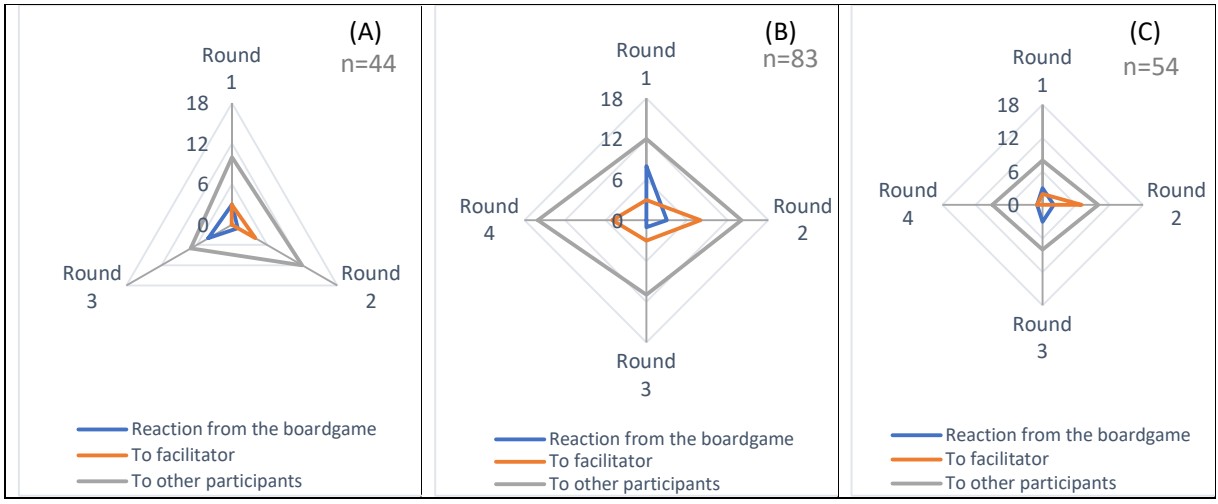

**Figure 6.** The summary of the direction of the sentiments extracted during the game sessions in Nanyuki (A), Teleswani (B), and Timau (C).

#### 2.6.2. Communication patterns during gameplay

The sentiments were further analyzed to identify their 'subtractive' and 'constructive' characteristics using the set criteria in Table 1. The results from the three game sessions reveal patterns of communication emerging in the different game rounds. Subtractive dynamics seem to reduce in the successive game rounds (Fig 7A, 7C, and 7E). To some extent, the scores for conflict and selfishness appeared to reduce to zero, especially in the final round. However, tension during gameplay remained relatively high and in some cases had a reversal as demonstrated in Fig 7C, and 7E. An increasing trend for constructive dynamics was observed especially with the increase in knowledge gain and the increased use of plural pronouns (Fig 7B, 7D, 7F). Collaboration and cooperation had the lowest scores among the constructive dynamics in the different game rounds. One important finding from this analysis is that even with a sudden increase in tension and conflicts (i.e. scores approaching -5), knowledge gain maintained to continuously increase throughout the different game rounds.

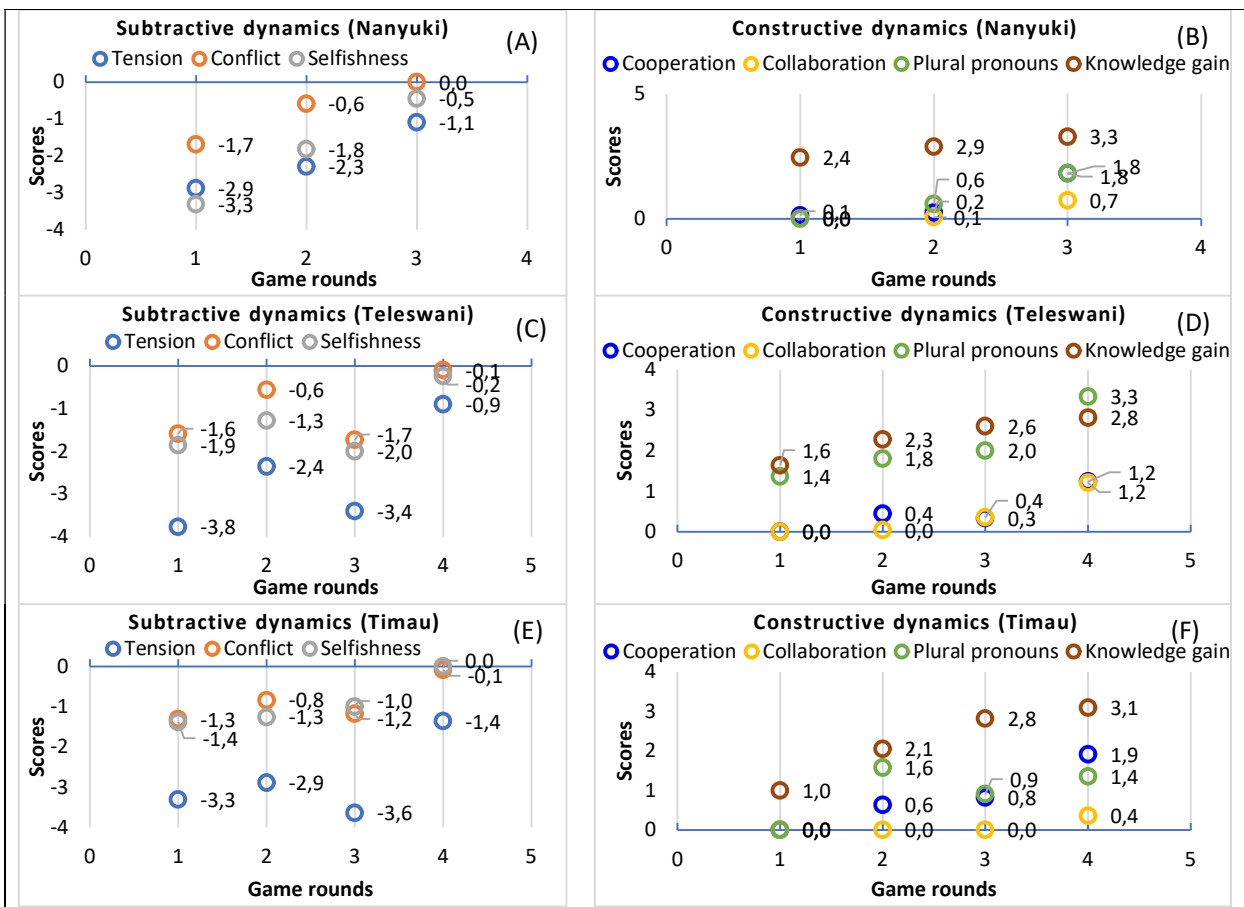

**Figure 7.** The scoring outcomes for the subtractive and constructive dynamics for the three game sessions.

The results of different game rounds were analyzed by plotting the total scores in Fig 7 in scatter graphs with four

quadrants (Fig 8). This helped evaluate the scores of a game round in relation to tension, conflicts, selfishness (i.e.

subtractive dynamics) and/or cooperation, collaboration, increase in knowledge gain, and use of plural pronouns

(i.e. constructive dynamics). A game round in quadrant 1, means the sentiments raised during gameplay reveal

higher characteristics of both subtractive and constructive dynamics. A game round in quadrant 2 has high

constructive dynamics and low subtractive dynamics. Quadrant 3 revealed high subtractive dynamics and low

constructive dynamics. A game round in quadrant 4 means low characteristics of both subtractive and constructive

dynamics. Apart from the Nanyuki game session where the first round was plotted in quadrant 3, all the game

rounds in the three sessions were plotted in quadrant 4. These results show that the game sessions did not reveal

extreme levels of both the subtractive and constructive dynamics.

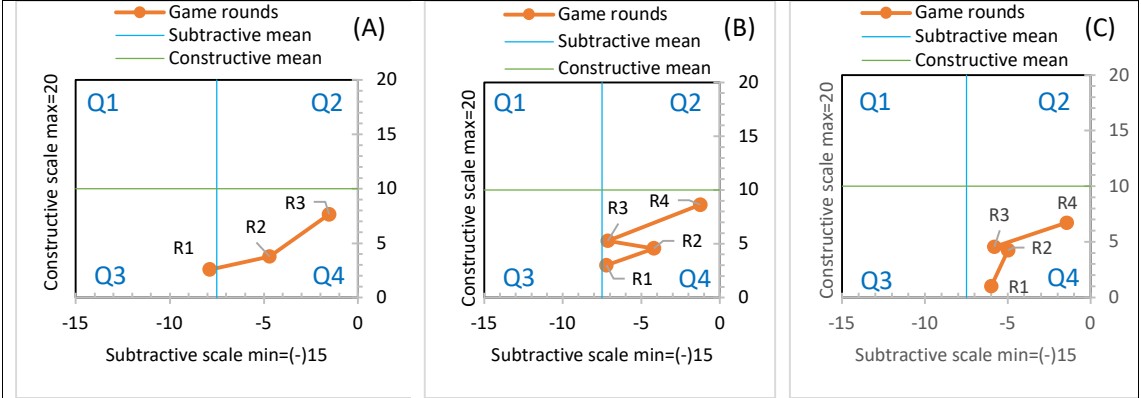

**Figure 8.** The status of game rounds, based on total scores (i.e. Summation of subtractive scores min = -15 and constructive scores, max = 20). For game sessions in Nanyuki (A), Teleswani (B), and Timau (C). A game round ending up in quadrant 2 would be desirable, as this would indicate increased cooperation, collaboration, knowledge gain, and plural pronouns, and at the same time a reduction in tension, conflicts, and selfishness.

One key observation with this type of analysis is that succeeding game rounds revealed a clear pattern toward

quadrant 2, mainly due to the constant increase in constructive dynamics. However, the subtractive dynamics revealed an oscillation pattern (i.e. an increase in one game round and a decrease in another game round) (Fig 8B and 8C). Although the game sessions had a few rounds of up to four rounds, this type of analysis helps shed more light on the engagement of stakeholders and their experiences during a game session. The pattern of a constant increase of constructive dynamics was further emphasized by plotting the subtractive and constructive dynamics

against the major decisions made by game participants (Fig S4). The results show that the oscillations on the subtractive dynamics in each round could only see a delay in the change of constructive dynamics but did not reverse the gains.

## 2.7. The solution space and game results

The solution space was used to plot the actual game results to help in the interpretation of emerging patterns. The

460 actual game results on net profits for upstream and midstream zones seem to lie in the upper limits of the solution space, compared to the pastoralist players in the downstream zone (Fig 9A, 9B, and Fig 9C). Plotting the game session results in the game solution space, showed that investment in agricultural expansion in the midstream zone may not necessarily lead to an increase in net profit (Fig 9B).

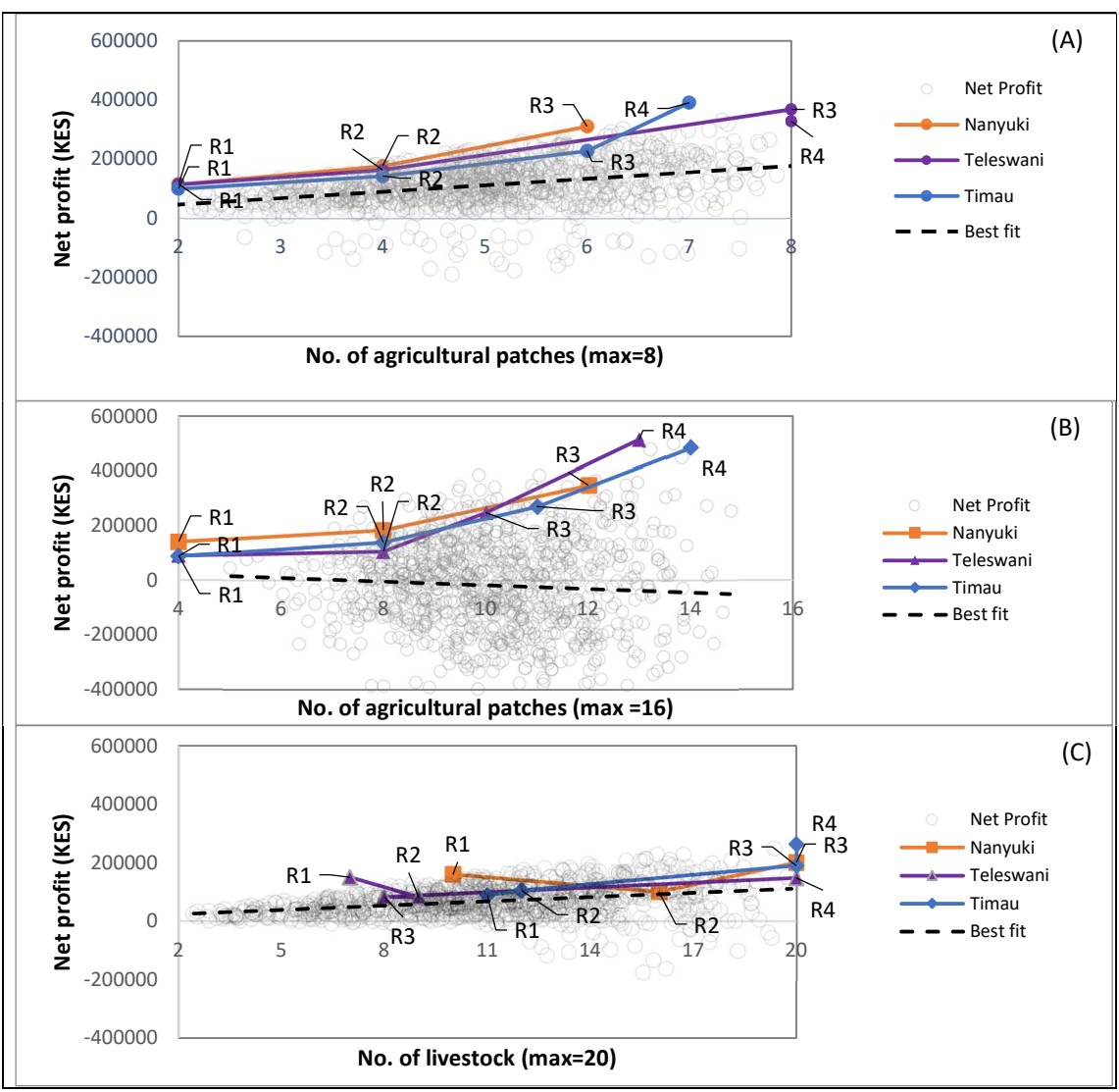

**Figure 9.** The board game economics in the upstream (A), midstream (B), and downstream (C) zones of the board game.

The agricultural expansion in the upstream and midstream zones of the board game was one of the major contributors to river water extraction during gameplay. The results show that water availability decreased with an increase in agricultural expansion (Fig 10). However, the actual game results showed a reverse of this trend, where water availability increased in Teleswani and Timau game sessions especially toward the final game rounds (Fig 10). A game round in Q2 in Fig 10 would be preferable as it means increased water availability despite the extreme agricultural expansion in the board game landscape.

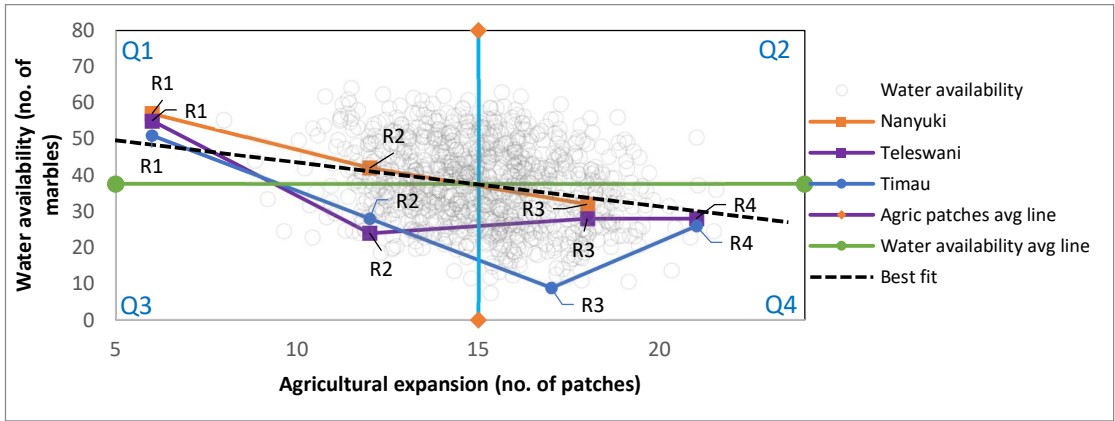

**Figure 10.** A solution space that links agricultural expansion and water availability on the board game.

The results reveal that an increase in water availability toward the final game rounds seems to coincide with increasing water harvesting decisions made by game participants (Fig 11A). Figure 11B compares actual game results that incorporated water harvesting decisions and projected game results assuming no water harvesting decisions were made. The results showed that by the end of the game session, water availability increased by 59%, 91%, and 50% in Nanyuki, Teleswani, and Timau respectively. This sends a strong emphasis on the importance of water harvesting decisions on the actual water availability against increasing agricultural expansion and livestock units during gameplay.

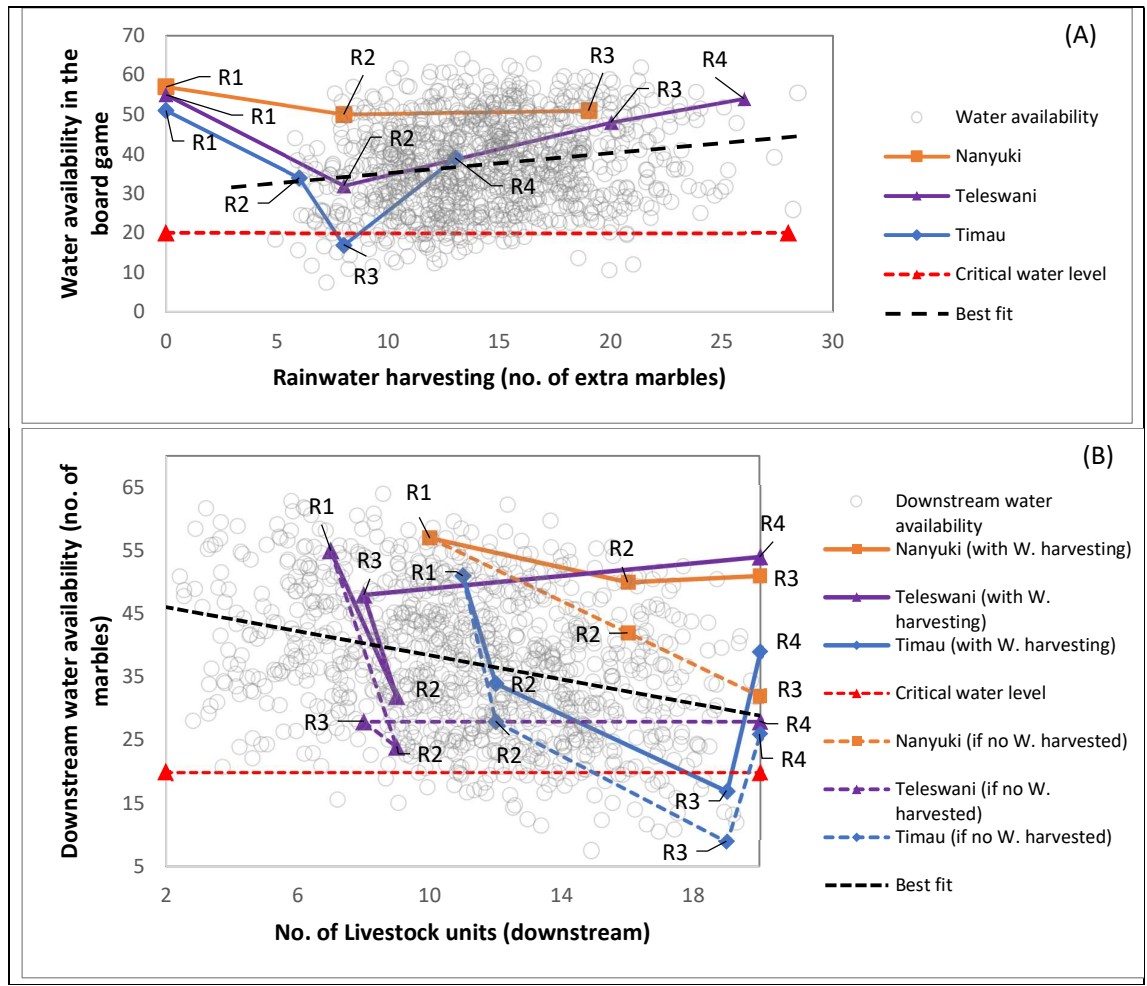

**Figure 11.** A solution that shows the effect of rainwater harvesting on water availability in the board game. General water availability in the board game (A), and downstream water availability (B). Actual game rounds with rainwater harvesting are shown with solid lines. Dotted lines are projected game rounds if no rainwater harvesting decision was made. In the ENGAGE game, the critical water level was assumed to occur when water availability goes below 20 marbles. This is because of the likelihood of livestock units in the downstream zone reaching the maximum of 20 (given that each livestock unit requires one marble). Hence the critical point where conflicts would occur regardless of other decisions made during gameplay.

## 3. Discussion

This study aimed to assess the potential role of the ENGAGE game in strengthening stakeholder engagement toward addressing complex human-water challenges of a river catchment. The study designed and implemented the ENGAGE game that mimicked the dynamics observed during the dry seasons in the upper Ewaso Ng'iro catchment. We assessed the type of decisions made during gameplay, the communication dynamics, active participation, and the implication of decisions made on water availability. Overall, the results show that implementation of the ENGAGE game as implemented in this study revealed the potential to strengthen stakeholders' engagement and shared understanding through stimulating active participation, increasing knowledge gain, and collectiveness, and minimizing individual interests and conflicts among game participants.

### 3.1.    Decisions made during gameplay

Based on the decisions made, upstream participants were observed to start by opening a few agricultural patches in the initial rounds, but this increased in the succeeding rounds, consequently increasing the demand for river water. This systematic approach could be attributed to equal capital distribution (i.e. money allocated to players) at the start of the game. However, as more profits were realized, we observed increased agricultural land expansion. Plotting of actual game results on net profits in the solution space revealed maximum profits for the upstream and midstream players, compared to the pastoralist players in the downstream zone. In addition, the solution space revealed that investment in agricultural expansion in the midstream zone may not necessarily lead to an increase in net profit. This can be attributed to higher costs for farming activities in the midstream zone. For instance, since the midstream zone is relatively dry, players are required to invest in two marbles per agricultural patch (for irrigation) compared to one marble in the upstream zone. Additionally, they are vulnerable to making losses due to the immigration of livestock units from the downstream zone even with moderate decreases in downstream water availability.

The downstream participants could not sustain their livestock numbers within the board game system, especially during the initial game rounds. This was mainly due to decreasing water availability and negative reception by the upstream participants who opposed the movement of livestock upwards into the agricultural zone. However, as the game sessions advanced toward the final rounds, we observed an increase in the use of plural pronouns, knowledge gain, and active participation among players during gameplay. As a result, the need to find remedies to reverse human-water challenges experienced by players as a result of their decisions manifested toward the end of the game. For instance, the need to adopt rainwater harvesting as a strategy to reverse the negative consequences spontaneously emerged among the players. This concurs with other studies that reported that game participants revise their initial decisions as they focus on possible solutions for the challenges observed during gameplay. For instance, in a serious game designed to explore and understand the complexities of flood mitigation options in urban-rural catchments, Khoury et al. (2018) reported that 70% of the game participants changed their initial decisions, and the best solutions were observed at the end of their game sessions.

This study provides a good example that demonstrates how streamflow variation in a river network connects the livelihoods of different communities within a catchment. For instance, the uncertainty of profits observed among downstream participants revealed how pastoralist livelihoods are affected as a result of the upstream river water abstraction activities. Water scarcity has been reported to be the main trigger of system instability during the dry season in the Upper Ewaso Ng'iri catchment (Gichuki, 2006; Kiteme, 2020). Nevertheless, besides the water

scarcity during the dry season, pastoralists also have to deal with pasture availability, hence an migration to humid zones in search of both pasture and water (Gichuki, 2004; The star, 2023). This study further reveals that in the context of agricultural land expansion, the situation gets worse for pastoralists who are in the transition zone (i.e.

between the agricultural zone and pastoralist zone). This is because they face pressure from other pastoralists who immigrate from far downstream in search of water and pasture while their upward migration faces inflexible resistance from the agricultural communities in the upslopes. This can explain the manifestation of fatal seasonal conflicts observed between upstream and downstream communities in the Upper Ewaso Ng'iro catchment (Kiteme, 2020; Lesrima, 2019). Within the game environment, higher profits were realized towards the final

rounds when the boardgame system was relatively stable. This is after significant investments in rainwater harvesting by all players during gameplay.

### 3.2.    Engagement of stakeholders

The ENGAGE game as implemented in this study was observed to stimulate and maintain active participation among players throughout the gaming sessions. Increased active participation in the game experimental

environment has also been reported in the literature (Jääskä and Aaltonen, 2022; Riivari et al., 2021; Speelman et al., 2014b). Maintaining active participation among targeted stakeholders is key to promoting local solutions to complex catchment challenges (Lim et al., 2022; Stosch et al., 2022). Similarly, this study observed relatively fewer questions and sentiments directed to the facilitator. This concurs with other studies that opined that scientists' role in a game environment is mainly to moderate the gaming session, hence, commonly referred to as a facilitator

(Javed and Kohda, 2020; Taylor, 2014). This indicates that with sufficient briefing and pre-game trials, the stakeholders can independently engage one another as they confront the human-water challenges presented in the game environment. We think that the ENGAGE game in its current form is easily transferable to other game facilitators working or interested in forested water towers and arid environments.

Based on sentiments raised by players, conflict, and selfishness appeared to decrease in succeeding game rounds.

However, tension during gameplay remained relatively high, and in some cases, there was a reverse trend. Tension and annoyance are common experiences in serious games where players compete during gameplay (Cidota et al., 2016; Oksanen, 2013; Poels et al., 2007). One important finding is that even with a sudden increase in tension and conflicts, knowledge gain maintains a continuous increase throughout the different game rounds. This indicates the strength of a gaming approach in increasing the knowledge among game participants. In addition, the sustained

increase of constructive dynamics amid oscillations of subtractive dynamics reveals the experiences within a game environment, for instance, the stakeholders can have a back-and-forth experience of selfishness, tension, and

conflicts, but at the same time attain a persistent increase of constructive dynamics. We therefore argue that the gaming approach can be seen to improve stakeholders' engagement leading to a persistent increase in the build-up of knowledge gain and the use of plural pronouns. We also recognize that it is likely that the perceptions by game

participants of the game facilitator matter, and the 'priming' with pre-game information and relational clues can influence game results, but this is a topic beyond this study and may need to be further explored.

The WRUAs have been in existence for over 30 years employing conventional participatory approaches to engage catchment stakeholders, especially in minimizing human-water related tensions and conflicts. This study shows that the first two 'issue cycle' steps, agenda setting (acknowledging that there is a problem) and shared

understanding of its causes and consequences, were readily addressed by the game in all three game sessions. As an alternative to existing conventional approaches, WRUAs can readily adopt the ENGAGE approach to engage catchment stakeholders in minimizing conflicts, promoting collectiveness and dialogues through active participation, increase knowledge on human-water interactions. The next steps on commitment to goals and means of implementation would depend on the way the game is made part of a longer-term process of interactions.

Collaboration and cooperation were observed to have relatively minimal increases which can be attributed to the nature of constructive dynamics. Knowledge gain and the use of plural pronouns can have a stronger individual bearing, compared to collaboration and cooperation which have a strong system-based bearing (i.e. inclined to the processes in the system). For instance, the sustained tension throughout the game rounds could mean players have limited interest in cooperating with the existing rules or collaborating with others.

The results in this study seem to contradict other studies that have reported that a serious gaming approach reduces tension, and increases cooperation and collaboration among game participants during gameplay (Morschheuser et al., 2017; van Peppen et al., 2022; Wendel et al., 2020). However, the notable difference between our study and other studies is game design and mechanics. An increase in cooperation and collaboration would directly manifest in a game design that intentionally forces players to cooperate and collaborate during gameplay (Wang and Huang,

2021). The ENGAGE game promoted independent decisions among players, in this case as they seek to amass huge profits. This explains the sustained tension during gameplay as players competed to attain individual goals. The assumption was that the players would learn from how individual decisions affect the system and thereby organically provoke a solutions-seeking attitude during gameplay. Therefore, collaboration and cooperation, in this case, manifest as a result of learning through the game process and are not embedded as part of key game

mechanics.

### 3.3.    Implications on water availability

The gaming approach can be seen to strengthen stakeholders' engagement and guide decisions toward addressing the complex human water challenges presented in the board game. The use of fines was one of the external motivations for pro-environment behavior in the ENGAGE game. The game participants were penalized for excessive water abstractions and the manifestation of re-imagination as one of the levels of internalization during gameplay (van Noordwijk et al., 2023), where participants felt peer pressure to regulate their ecological footprint to reduce the impact on the system. There was a decline in water availability in the initial game rounds and then an increase toward the final game rounds. This could be attributed to rainwater harvesting decisions made by game participants in the final game rounds which saw water availability increasing between 50% and 91% in the three game sessions. Therefore, despite increases in agricultural land expansion and livestock units, guiding stakeholders to practically understand the importance of rainwater harvesting and storage is an important factor for water availability in the river catchment. This finding concurs with other studies that opine that rainwater harvesting can be a feasible solution for water scarcity problems during dry seasons (Irshad et al., 2007; Velasco-Muñoz et al., 2019). However, these are supply-side solutions which can readily fit the context of water scarcity problems where there is urgent need to provide 'additional water' to quell existing conflicts as conceptualized in this study. Overall, there could be thresholds to the extent to which the supply-side solutions are sustainable. Increasing water storage capacity, despite improving water supply can increase stakeholders' vulnerability due to reservoir effect (Di Baldassarre et al., 2018), human displacement (Asmal, 2000; Kuil et al., 2018), inefficiency due to excess water 'reallocation' (Kuil et al., 2018), among other anticipated effects.

The ENGAGE game showed to be able to guide catchment stakeholder discussions involving small-scale farmers, pastoralists, water resources managers, etc. toward addressing the water availability challenges of a river catchment. Gaming can be an important practical tool that can be used by river-based organizations (RBOs) to increase understanding among catchment stakeholders. Adoption of a gaming approach in existing participatory approaches as often used for implementing IWRM, can enhance participation and allow stakeholders to directly interact with the 'wicked' problem, testing scenarios in decision-making, and social learning for collective action. This can lead to the production of informed IWRM outputs such as farm and catchment management plans. The challenge for serious gamers is the extent to which the gaming lessons can be scaled out beyond the game experiment environment to cover a larger population of a river catchment.

**3.4.    Limitations of the study**

Despite the promising results of this gaming approach, we highlight a few limitations of this study that may have led to bias in the study results. First of all, the ENGAGE game sessions can only accommodate ten participants at a time. This means the results can only be interpreted at the experimental stage and more time and resources may be needed to cover a larger sample of catchment stakeholders. Given the time and resources, the study conducted three pilot-game sessions with the maximum number of game rounds being set to four. The assumption was that Phase 1 consisting of two to three rounds was sufficient to expose players to the system dynamics, and a final round would allow for reflection on possible solutions. We argue that conducting several game rounds without subjecting the participants to a final reflection round may be an alternative way to assess the emergence of gaming outcomes. Without a reflection round, we speculate some game sessions would have ended with worst case scenarios, which still would have been important results as an emphasis of 'wicked' nature of human-water challenges. Although this would require a larger number of game sessions to gather a sufficiently large sample size of game sessions. Assessing game outcomes from such a larger sample size may be desirable to study more patterns and more emergent patterns of this gaming approach.

In this study, the communication analyses focused on sentiments related to human-water issues expressed by the game participants in each game round. A game round took an average of 15 to 20 minutes. Non-verbal aspects of communication such as intonation, pitch, tempo, and cues such as posture and gesture (Duncan, 1969; Gozalova et al., 2016) were not directly accounted for in this study. However, during the video analyses, the scoring considered some of the qualitative aspects in determining the scores. For instance, a sentiment such as *"You can't settle here, the river is already dry here"* when said by a participant at a lower pitch, and seated, may indicate there is tension but the scores for level of conflict may be lower than when said by a participant with visible physical gestures such as standing up, pointing a finger to a participant, blocking a participant from completing a particular task, etc. Besides, this study did not consider things such as the personality differences between the participants, or their real-world relationships. Some participants may be shyer than others during gameplay, due to such factors.

Nevertheless, the results of this study can contribute to the body of knowledge on using the serious gaming approach to address complex human-water problems. Therefore, despite the small sample size, the results in this study can be used to inform human-water policies and modification of water allocation rules at a river catchment. Serious gaming presents an opportunity for stakeholders to interact with the 'wicked' socio-hydrological problem and guide stakeholder discussions on water management. The game environment allows for real-time reflection

through the creation of a fictional setting and a common pool for the stakeholders to explore decisions and impacts simultaneously. This creates the opportunity to change from one's usual position and see the wider picture in a safe environment. This is different from the real-world situation in which a blame game exists of the 'person living upslope as the contributor' to the water problem experienced downstream as noted in (Wamucii et al., 2023). Furthermore, as reported in other studies the ENGAGE game also showed the good properties of simulation games

mentioned in the literature for motivating the intentions of the stakeholders toward sustainable behaviors (Bassanelli et al., 2022; Douglas and Brauer, 2021; Hirsch et al., 2010; Lieberoth et al., 2018). Some of the unique qualities relatable to the ENGAGE game as reported in literature include; universal appeal, the ability to capture and retain participants' imaginations and intentions, simulation of near reality, immediacy, interactivity, and visual feedback (Fox et al., 2020; Sajjadi et al., 2022; Schuller et al., 2013; Wolf, 2020). These qualities enable game

participants to interpret, relate, argue, criticize, investigate, and construct new knowledge – hence the manifestation of pro-environmental behaviors (Fox et al., 2020; Sajjadi et al., 2022). This study observed that the board game allowed participants to reflect on complex human-water issues in the catchment such as; water scarcity, downstream-upstream conflicts, economic losses, rainfall uncertainty, crop failure, harsh national government rules, and fines, etc.

Additionally, since each participant in the game has a role to play, they have a stake during the game. They tend to be active in pursuing their stake while focusing on their roles. This is different from other participatory methodologies where there is a claim to engage participants e.g. through group discussions or dividing the participants into random small group discussions (Global water partnership 2005; Voinov and Gaddis 2008; Hare 2011). This often results in some members less actively contributing to group activities (The university of

Queensland, 2023; Burke, 2011), which is mainly due to power differentials among stakeholders (Daré et al., 2018). In a serious gaming environment, all participants are engaged and glued to their specific roles and they do not necessarily have come to a common position. The end outcomes in a game environment depend on individual experiences in the different game rounds.

## 4. Conclusion

This study assessed the potential role of serious gaming in guiding community discussions on water-related challenges. We did this by creating a game, an ENGAGE game, and playing three sessions with farmers and other stakeholders from the target region. Overall, game players participated actively, gained knowledge, acquired a sense of collectiveness, and minimized conflicts. The subtractive dynamic oscillations especially due to tension during gameplay could only see a delay in the change of constructive dynamics but did not reverse the overall

gains. Even under complex human-water challenges under expanding agricultural lands in the upstream and midstream zones and increasing number of livestock units in the downstream zone, possible solutions can easily be explored in a gaming environment. For instance, it was observed that there are possibilities of reversing the dry season water scarcity problem in the river catchment through rainwater harvesting. Whether the results realized in a gaming environment could impact reality is still unclear. Post-game discussion revealed positivity on the usefulness of the gaming approach in promoting sustainable behaviors, where game participants indicated willingeness to adopt a few lessons from the game. However, it may require more game sessions and long-term studies to assess the impact of serious games on strengthening stakeholders' engagement and maintaining sustainable behaviors in real life. Nevertheless, the results of this study have important implications for water management and can be used to inform human-water policies and modification of water allocation rules at a river catchment. Water resources management stakeholders can work with the ENGAGE game as a starting point for catchments with similar context as Upper Ewaso Ng'iro basin.

**Availability of Data and Materials** Data and materials can be made available by contacting the first author.

**Supplementary Information** The online version contains supplementary material available at https://doi.org/10.18174/638544.

**Author Contributions Charles Nduhiu Wamucii:** Study conceptualization, Data collection, Formal analysis, Methodology, Writing – original draft. **Pieter R. van Oel, Adriaan J. Teuling, Arend Ligtenberg, John Mwangi Gathenya, Gert Jan Hofstede, and Meine van Noordwijk**: Contributed to the analysis of the results and reflection of the study results. **Erika N. Speelman:** Guided on game conceptualization, Methodological design, Data collection and critically reflected on the analysis and interpretation.

**Funding** This research was made possible by Wageningen University through the Scenario Evaluation for Sustainable Agro-forestry Management (SESAM) project that was funded by its Interdisciplinary Research and Education Fund (INREF).

**Conflict of Interest** At least one of the (co-)authors is a member of the editorial board of Hydrology and Earth System Sciences.

**Acknowledgments** The authors would like to extend gratitude to the households that participated in the game sessions. We also acknowledge the respective water Resources Users Associations (WRUAs) for supporting the first author in the mobilization of game participants during field data collection.

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
