# Peer review of "Guiding community discussions on human-water challenges by serious gaming in the upper Ewaso Ng'iro river basin, Kenya"

_EGUsphere, 2023_

## Author Comment (AC1)

**CC1**: **'Comment on egusphere-2023-2459', Wim Douven, 14 Dec 2023**

Dear Charles and team, I enjoyed reading your manuscript. I find the topic very interesting, and it is good to have another study on serious games. Below some comments, questions, that might be helpful in finalising the manuscript. I am happy to hear you enjoyed reading the manuscript. Thank you for the positive remarks on the manuscript and constructive suggestions and comments.

**Comment 1:** In the section on IWRM (line 66 and further), you mention that workshops do not promote different experts to engage with wicked problems. But doesn't that depend on how you design such workshops, like how you design serious games ..? Thank you for the question.

**Response:** Yes, we agree that that there are many different participatory methods developed especially to promote the engagement of 'local experts'. One of the most familiar is focussed groups, but many others are around including Dramas. Designing serious game sessions may differ from designing workshop sessions. Firstly, serious games are co-designed where both the 'scientists' and 'local experts' are involved in the development of serious games (way before actual game sessions). In the workshop sessions, the scientists' are the most active persons, e.g. when making PowerPoint presentations, explaining an idea, reporting case studies, or simulating a model and its possible results. The participation of 'local experts' is relatively 'passive' in the workshop sessions e.g. sitting and listening to presentations made by scientists. Some workshops may divide participants into small groups for discussions, experimenting, and demonstrating workshop issues. But in most cases, the group discussions tend to have active and less active persons - *in Ln 547 of the manuscript, we have argued this could be due to power differentials among participants...* Workshops often are not the right way to deal with wicked problems.

There is a difference in the game sessions. Serious games provide a 'buffer' to mitigate powerplay and hierarchy, make it more easy for participants to express more freely, since it is 'just' a game. Serious games game provide a boundary object commonly not found in workshops. The 'scientists' are less active, and tend to play the passive role of facilitating the process (especially at the start of the game rounds), while participants are the most active persons (in small groups) right from the start to the end of the sessions. The facilitation role tends to decrease as the game progresses, and at some point 'scientists' become the 'observers' of what is happening in the different small groups of gameplay. Scientists tend to learn more this way, as opposed to when they are the main presenters in conventional workshops. The game participants are vigorous as they interact directly with 'virtual systems' such as Board games and also actively engage one another (e.g. disagreeing, setting rules, seeking alternative strategies, etc), the scientists do not control the actual gameplay. *In Ln 542 of the manuscript, we have argued that '…..since each participant in the game has a role to play, they have a stake during the game. They tend to be active in pursuing their stake while focusing on their roles…'*

We agree that if we design workshops similarly to game processes (including having game-like engagements during small group discussions), then designs of workshops and game sessions could both promote different experts to engage in a meaningful way. Still, in most conventional workshops the 'scientists' are the main 'active persons' while 'local experts' are the 'passive' participants. By design game sessions reverse these roles.

**Comment 2:** I note that the lit. review on games is overall very positive on its effects on finding joint solutions etc.., are there no limitations and challenges reported in literature ? You mention one in your discussion in line 520. Did you come across other limitations, challenges in the sessions? Thank you for the questions.

**Response:** This study focussed on the literature that has attempted to report on the 'effectiveness of gaming approach'. Hence the purpose of this study was to 'investigate these opinions/claims' by assessing the potential role of serious gaming in strengthening stakeholder engagement. This study contributes to the debate on the 'effectiveness of serious gaming' including how some results contradict past studies see Ln 466-476.

We have highlighted a few limitations in this study (given the focus of the manuscript) in Ln 500 – 521. Definitely, we can probe more about the limitations and challenges of serious games (in other general/specific game settings) and incorporate them in the discussion section (limitations sub-section of this manuscript). Generally, among the limitations/challenges, the most important one is: you cannot play

with people if they are in the mood to fight. So if there already is a conflict, that needs to be dealt with first. Another complication can be that some people have their reservations to engage in a game session if particular other individuals also participate - for instance, bosses with employees, in a hierarchical society.

**Comment 3:** In game conceptualisation (Section 2.2), it might help the reader to give a brief summary about sessions, rounds, and phases, and how they relate to the catchment / sub-catchments. Thank you for the suggestion.

**Response:** During this stage of game conceptualization, there was no playing the games per se, but this is the 'setting stage', which involved bringing all possible ideas to the table, to help in crafting a serious game that fits the context of the case study area. The ideas were gathered from past studies, through the ARDI approach, and conducting community discussions with communities in three sub-catchments. Game testing sessions are described in sub-section 2.3, where feedback was key to help in refining the final game.

**Comment 4:** Data collection (Section 2.4) is there after the step 'modeling the game solution space' not also a step where you collect data about discussion and feed-back, which might result in adjustments of positions and solutions .? As it is an interactive process, it is collecting data and providing data, right? (so feed-back, and giving space for learning)? Thank you for the questions.

**Response**: The data collection description in sub-section 2.4 focussed on data collection based on the final ENGAGE game_v1 described in Box 1 and Supplement 1. To ensure comparability of data across different game sessions, no further adjustment was made and v1 of the game was maintained (as developed) in the three sub-catchments. Yes, there is a feedback session at the end of playing the games, where participants are allowed to give their feedback and key lessons on the game sessions. The qualitative feedback was useful in crafting some of the reflections included in the discussion section e.g. in Ln 523 ... the *game environment allows for real-time reflection through the creation of a fictional setting and a common pool for the stakeholders to explore decisions and impacts simultaneously'*......in Ln 542 .......'*each participant in the game has a role to play, they have a stake during the game. They tend to be active in pursuing their stake while focusing on their roles'*.......

This manuscript focusses on the 'decisions made and qualitative aspects of gameplay' i.e. sentiments raised during the actual gameplay. It is paramount to mention that "*another manuscript (also reporting on 5 additional game sessions*" focusses on the experiential learning of ENGAGE game_V1, and how stakeholders' perspectives change before and after game sessions, including the post-game individual interviews with different game participants is about to be submitted.

**Comment 5:** An important discussion point you touch upon is how the game results translate to real life situation (line 560 and further). I wondered was this discussed in the sessions, at the end? How the game was valued, also for real-life? That could already give some initial insights? Thank you for the questions.

**Response:** Yes, a conversation on translating game experience to real-life situations was a key discussion point at the end of the game sessions. There was high positivity on the usefulness of the gaming approach in promoting sustainable behaviors, and all game participants indicated they would adopt a few lessons from the game. The following are some of the direct quotes from some of the game participants:

- *"I need to reduce dependency on the river, I need to construct a small dam"*
- *"I need to manage the land use better, considering the wet and dry seasons. I also need to construct a small dam"*
- *"I need to stop clearing the bushes and forest for agricultural land expansion, I better manage what I already have"*

However, this manuscript recommends further follow-ups to assess the impact of serious games on strengthening stakeholders' engagement and maintaining sustainable behaviors in real life.

Follow-up data collection was done in another study (*"The Manuscript on how stakeholders' perspectives change before and after serious games"),* whereby game participants were traced 5-7 months after the game sessions, and most of these aspects (game experience – real life situation) have been explored.

---

## Author Comment (AC3)

**Comment by Jean-Philippe Venot**

This is a paper that investigates the use of a serious game to discuss issues of water management at watershed level in Kenya. I enjoyed reading the paper; a lot of work has gone into the design, implementation and analyze of the research. I think this type of paper reflecting on the use and scope of such participatory approaches for improving water governance is important.

That being said, I feel the paper shares what I see as a bias in the existing literature on games for sustainable natural resources management, that is, a rather positive outlook on the changes/transformations these games may trigger and a focus on their promises and potentials – that is only partly supported by what is actually described in the papers (that is what happens during "game sessions" as opposed to "in the real world").

*Response: Dear referee, thank you for the acknowledgment of the relevance of our research exploring the potential of participatory approaches such as serious games for improving water resources management and governance. We plan to address your major five comments and specific comments as shown below:*

(1) **MAJOR COMMENT 1: Better clarify how the process they steered was fundamentally different (or not) from other more classic participatory approaches notably in relation to co-production of the tool (as opposed to a contrasting active game with passive attendance to workshop).**

Line 66 and following: The authors attribute some of the shortcomings of IWRM to the fact that IWRM projects and policies make use of "classic" participatory approaches that would not really allow different actors to engage with the "wicked problem" at hand. In other words, participation in IWRM is not real participation. This is a well-known critique of IWRM, which I share in part, but I think there is a danger of building a strawman for the wrong reasons here, just to make the point that games do not share these shortcomings. I agree when the authors answer the previous community reviewer that games allow a different, more active, mode of engagement than classic workshop based that use power point presentation. But there are many ways to conduct workshop that can provide meaningful engagement of participants: participatory mapping, experimentation with art-based visual, etc. are some of these options.

I think the issue is less about the modalities of facilitation of workshops (and the tools used) than how multiplicity of knowledge is handled and how/who decides on the terms of the debate. Notably, who ends up deciding what the 'wicked problem' is. Games can actually reify the 'wicked problem' on the basis of an outsider/expert view; if so I would argue that the nature of engagement of participants remains as limited as in other approach or at least is constrained in major ways.

What bothers me is that the phrasing of this paragraph (and many others) seems to indicate that the authors hold the view that there is ONE "wicked problem" (likely a lack of coordination between downstream and upstream users). It makes it seem as if this problem existed almost independently of the people who participate, being identified à priori by a subset of actors (though there is scope for refining the problem at hand through the use of the game itself). If that is the case, then, the game developed is likely to have very similar shortcomings than "classic" IWRM based approach.

I think the critique of IWRM approaches would be more powerful and convincing if it were to be done at the level of how are the terms of the debate/participation set and by whom (as opposed to looking at the specifics of interactions and their presumed passivity), and then to show, how this particular study did something different, hence created the room for another type of engagement by the people who participated in the game session. The point here is to pay a bit more attention to the politics that framed the entire research process and operated silently in the background as opposed to take the dynamism of the game sessions as a sign of pro-active engagement and changes to come in the watershed. The authors may want to look at https://hess.copernicus.org/preprints/hess-2023-

164/ currently under review in the same journal. This paper reflects on some of those issues around the practices of hydrological modeling not games, but I think there are many similariries.

I do not doubt the authors have done something quite different from "classic passive workshops"; my qualm is with the way authors describe the differences. Instead of making clear what the differences were, the way it is written points to fundamental similarities in the assumptions underlying the approach (which may not have been there). I feel clarifying this point through some reformulation is important and would strengthen the argument of the paper.

**Response:** *Thank you for the detailed comments. We agree with the referee that the lack of 'real participation' is a well-known critique on IWRM. We are happy to see the referee agreeing with our response to the community comment that serious gaming is a different, more active, mode of engagement than classic workshops. Interestingly, the reviewer also confirms that there are many ways of conducting workshops that can provide meaningful engagement of participants (particularly the inclusion of 'hands-on' interactive activities in the workshops e.g. participatory mapping, experimentation with art-based visuals, etc). This re-affirms our arguments on the need for 'hands-on interactive activities' in the conventional participatory approaches. And not just the inclusion of 'hands-on interactive activities', but those activities should yield results that can inform or indicate how people interact with their environment and opportunities for collectiveness and collaboration. The fact that there are no uniformity/standard ways of conducting participatory engagements in IWRM demonstrates the need to develop and evaluate different participatory approaches. Serious gaming is one of the possible strategies to be investigated more deeply given the growing complexity of human-environmental issues.*

*Given the context of the case study area, where there have been water-related conflicts annually (even fatal conflicts), and the fact that under normal circumstances, the downstream communities do not sit or see eye to eye with the upstream communities **(see Ln 121-124)**; developing and testing an alternative participatory approach such as a serious game is timely. This is especially so since, the River Basin Organisations (i.e. WRUAs) in the case study area have been in existence for over 30 years now, and mostly the communities have been exposed to the conventional engagement approaches e.g. workshops, community discussions, group discussions etc **(See Ln 184 – 187).** And yet the situations are getting worse each year, with the national government 'reacting to calm the conflicts' but with no permanent solutions **(See Ln 189 – 198)**. We will revise the manuscript to make the contexts more clearer.*

*In conventional engagement approaches for IWRM, the leading scientists have developed hydrological models that are presented to the stakeholders/participants to shape human-water discussions. Different scenarios can be tested and results presented during a workshop to allow deliberations on the way forward towards resilience. This in some way has a sense of 'top-down strategy'. In a serious gaming process, all stakeholders (scientists/non-scientists/ including low-educated) define the problems, priorities, and test scenarios themselves, and 'revise' their perceptions based on experiences from playing the game with less 'scientists' input'. Sometimes, the results from a gaming experience can be different from 'scientists' expectations as game participants bring 'reality into playing the game' as opposed to a 'stylized model'. During game sessions, it has been very common to hear stakeholders link game results to events in real life (e.g. ….."oooh, could that be the reason, the river stopped flowing at point xxx"…..."I see the reason why the rivers these days are brown these days compared to clean water we experienced during our primary school days, soil erosion from expanded cropland areas…". The involvement of catchment stakeholders right from game conceptualization to implementation, reverses the engagement to a 'bottom-up strategy'. The 'gamification' component is an important aspect, in our own opinion, any model can be 'gamified' to allow 'local stakeholders' to learn and understand the 'scientific models work (but scientists must be willing to 'unlearn' to 'learn'). It is this increased system understanding that can inform realistic catchment management plans. We will revise the text to expound our argument on IWRM short comings and also reflect more on serious games short comings and opportunities.*

*Regarding the referee's comment that games can actually reify the 'wicked problem' on the basis of an outsider/expert view we argue is not the case with serious gaming. We argue that the outsider view is highly managed in the gaming sessions. **(See Ln 88)** The design of a serious game is an iterative process that evolves with the participative process whereby local stakeholders are actively involved in defining the 'wicked problem', and designing the questions, simulations, and outputs. During actual gameplay, the outsiders/experts are more 'facilitators' than the 'leads'. Hence the nature of engagement of participants is not as limited as in other conventional approaches e.g. workshops where scientists take the lead (see our responses to the community comment on this point).*

*We do not hold the view that there is ONE "wicked problem" in the case study area. We have actually defined what we mean by 'wicked problems' **(see Ln 47-51)**. Just like any other modelling logic, the game tries to simplify the representation of the socio-environmental dynamics. We have stated in **Box 1** that the ENGAGE game mimics the dynamics observed during the dry seasons in the upper Ewaso Ng'iro catchment. The downstream-upstream reactions result after the processes both in the socio and physical dynamics (it is not just a lack of coordination between upstream and downstream communities, but an array of possibilities or outcomes that can be observed with serious games).*

*We agree with the referee's comment that the critique of IWRM approaches would be more powerful and convincing if it were to be done at the level of how are the terms of the debate/participation set and by whom and the politics that framed the entire research process. As mentioned earlier, the outsider has no exclusive power to dictate the process. We will revise the manuscript to make this important aspect of serious gaming. Right from the conceptualization of the game, the local stakeholders have been involved in defining the key actors, resources, dynamics, and interactions. And this is one of the ways that games differ from other conventional participatory approaches.*

*The pre-print shared by the referee ter Horst et al,. (2023), actually re-affirms the politics around conventional approaches and modelling where the 'outsider' e.g. a hydrological scientist/modeler is the one defining the model components depending on the area of interest. This has been the norm for years, and conventional models have been used to shape discussions on water resources management and governance. It is very common to have presentations of model results e.g. during a workshop. Most of these models are 'black-boxed' and most of the specifics are left to the modelers. The opposite is true in serious gaming, the whole process is 'open' and defined in collaboration with stakeholders (scientists and non-scientists) at all stages, conceptualization, refining, and implementation. In other words, you cannot present or 'force' a game, that participants have no clue about or are not interested in (the game must mimic the real socio-environmental context of the targeted stakeholders). We will revise the text to include a paragraph reflecting on the politics around conventional modelling and gaming, and expound on 'black-boxed' and 'open' approaches.*

*We however recognize advanced game versions that go beyond a physical board game or use of cards into developing an Agent Based Model. ABMs can be helpful in understanding the 'system dynamics' through computer simulations, however, in situations where the interest is to explore ways of peaceful coexistence between worrying communities due to hydrological dynamics (i.e. strengthening engagements), the option of using 'real agents'/real participants in a 'safe environment' remains the better option since stakeholders can express their interests, concerns, test scenarios and explore possible way forwards. Therefore, investigating the potential of serious games in the context of strengthening stakeholders remains useful, which can be seen as advances in the field of water resources management and governance.*

 (2) **MAJOR COMMENT 2: They could also engage a bit more with the risks/limitations of games that have been discussed in the literature**

Line 93 and following. The way to depicts serious games and what they can help achieve is overwhelmingly positive. I think this reflects rather well the existing literature on serious games, most of which is actually written by people involved in their design and implementation – which may make them a bit less likely to adopt a critical stance towards these tools. Yet, I think the authors could engage with some of the limits of games that have been discussed in the literature, notably by

members of the ComMod collective they are likely to know. There is some work in relation to conflicts and existing relations of domination (Barnaud & Van Paassen, 2013; Bécu et al., 2008; Mathevet et al., 2014), on the roles and postures of facilitators and modelers (Barreteau et al., 2003; Jonsson et al., 2007), or the risk of manipulation (Barnaud et al., 2008; Halbe et al., 2018). A recent overview recognizes ''potential biases" (Barreteau et al., 2021). See also (Jonsson et al., 2007) on whether games are "truly participatory". Also, the fact that games simplify complex problems (line 102) is presented as a positive thing (I suspect in relation to legibility of the said problem). This may be the case, but it also raises a lot of questions: who decides what is represented and what is not, what are the implications of those choices… In other words, what are the politics of the game itself? These questions ought to be at least mentioned I think.

*Response: Thank you for the comment. This observation on positive look on serious games was also raised in the community comment. I think the context of the study area where there are complex water-related tensions and conflicts, may have led to the language used sounding positive. Especially given that the River Basin Organisations have been in existence for over 30 years (engaging stakeholders using the conventional engagement approaches) and the passion for exploring alternatives to stakeholder engagement may sound overwhelmingly positive. This will be relooked and expounded so that the point is clear. We agree with the referee's comment regarding diving deeper into discussing limitations around serious gaming. We appreciate the sharing of specific literature that has delved into this discussion. We will provide a detailed review of the limitations of the gaming approach in the revised version, both in the introduction section and in reflecting game results further to accommodate reported game limitations.*

**(3) __MAJOR COMMENT 3: Provide a bit more information on the methodology (including the link between the game and the computer model)__**

I feel there is a need to clarify the methodology. Notably, in the first paragraph it is not clear to me what the "solution space" of the game (what are 'responses' to rules?). It is also not clear what the authors mean by "overall performance of the game results" (how is performance assessed, is it in relation to water availability/sharing? In relation to how active participation was…)? In relation to that, section 2.4.3 discusses a modeling of the 'game solution space' and mentions a 1000 runs but what are those runs? I assume the authors used both a physical game and a computer version of it and that they used the later a 1000 times generating random actions; this to situate the results of the specific game sessions in the "realm of possible". But I can only assume this: the relations between game and model need to be made explicit and the authors need to explain, why the use of the model was useful, what did it allow to do?

*Response: Thank you for the comment. In __Ln 263__ we have described the Solution Space as …..an envelope within which ENGAGE games operate, by understanding the minimum and maximum values of the various game metrics…...(i.e. the space of minimum to maximum of game results). The 'overall performance' mentioned in __Ln 137__, refers to actual game results plotted within the Solution Space. In other words, the general trends observed in the different rounds of the game. We will revise the wordings to give clarity on what we mean by the 'Solution Space'.*

**(4) __MAJOR COMMENT 4: The mechanics of the game in the core of the paper (fez readers are likely to read the supplementary material)__**

A lot of key information on the mechanics of the game (including how the connections between actors and water flows are conceptualized) are included in the supplementary material. I think only few readers are likely to read the supplementary material and it might be a good idea if the authors extended the presentation of the game in the core of the paper slightly. I'm also not sure the game should be presented in a "box". Normal section numbering would do. In terms of presenting the game, I think it would be interesting to have a sub-section describing the board and the players, another one describing briefly the different action/decision each actor can make and how this impact water flows (at a conceptual level; not entering in the details of the calibration), and another one on the "typical" process followed during each session – this is already in large part in the box. I think the

graph on the system dynamic modeling could support this presentation of the game, as opposed to be presented in relation to the "solution space" of the game. The supplementary file can then focus on the calibration aspect. Presenting the game as such would also allow having part of the discussion engaging with the key stage of game validation that is not really described at the moment (what hypothesis the authors made were validated, which one were not…? for instance deciding that water dries up from downstream when people abstract water from upstream is a strong choice. I would be curious to know if that triggered some discussion also because, materially, it involves taking a marble from one area of the board to put it in the other one, which is not really how water abstraction works in the real world.)

*Response: Thank you for the comment. This will be considered in the revision. We can expand the game description in Box 1. We also agree with the suggestion to remove the 'box' and have the game description in normal numbering and possible with sub-headings:*

*Gameplay description and process*

*1.1. Description of Boardgame and players*
*1.2. Key actions and key outcomes expected in the game*
*1.3. Potential impact on water resources availability*
*1.4. Possible reactions expected by actors and feedback*

*Regarding the comment on validation, this is described in **sub-section 2.3 (Game pre-testing and validation).** The type of validation described here is 'expert-based validation type' where the catchment stakeholders are involved to confirm the processes and game outcomes and assumptions made (i.e. legitimacy, credibility and salience of the ENGAGE game). The game tries to mimic real actions/reactions of actors during dry seasons, and the game results should also mimic the real observations which have to be confirmed/validated by the 'local stakeholders'. This is a long iterative process, involving several game trials, refining until 'key human-water issues' are well represented and polished. For instance what you have referred to as…….deciding that water dries up from downstream when people abstract water from upstream….. was not a decision made solely by the 'scientist'. But a collective reflection on how to represent the 'real processes' in the study catchment e.g. rivers drying from the bottom-upwards. This is because of excessive river water abstractions in the upstream zones. In this case, whenever a marble is taken out of the river, the facilitator pulls the sting to mimic reduced flows, which ends up with no flows in the downstream zone. Whenever a marble is taken out, it is assumed as 'a hydrological loss' (i.e. 'enhanced evaporation') due to higher atmospheric energy (PET) as you move from upslopes to downslopes (arid zones). Some of these technical explanations including how land conversion from natural vegetation to cropland reduced dry season flows are explained in the supplementary material.*

(5) **MAJOR COMMENT 5: Finally not only discuss the fact that the game seem to have enhanced communication among participants and their promises in general, but also issues related to the politics of the game themselves, that is, the extent to which the representation of the system was accepted/validated/discussed by participants and the scope for the game to change the broader rules of the game and conditions of operation of the WRUAs.**

The discussion focuses on what has happened during the game session and it is important. The game sessions are likely to have improved communication and cross learning among participants but to which extent did it actually provide ways to solve the structural issues faced by WRUA that are identified in line 188 (weak enforcement of policies/laws, water abstraction regulations, water metering requirements, protection of riparian corridors/forested areas, etc.)? Such structural issues are likely to relate to the institutional strength of the WUAs, how legitimate it is vis-à-vis other actors, the (human) and financial means it has… Even if it is not the objective of this paper to assess the scope of the game to change things in the real world, I think it is important for the authors to engage in a discussion around the promises and potential of games in relation to these structural hurdles.

Most of the literature on games tend to focus on the design process and on what is happening during game sessions, and that's why most of the literature is positive: there is a lot of interesting things happening, some of which do have transformative potential. But in doing so, the game and the overall research process is also "removed" from the overall political-economic context in which it is implemented while this context is pivotal for the promises of the game to actually materialize or not. While describing the overall political-economy that influences water management in the case study area is beyond the scope of this paper, I think it is important the authors reflect a bit on it and how this might influence the future of the game outcomes – to avoid falling in the trap of highlighting promises that may never materialize.

*Response: Thank you for the comment. The discussion strongly reflects on the potential of serious gaming as an alternative, given the context of the study area. We will revise the discussion to also reflect more on the WRUA structural issues. As mentioned above, the WRUAs have been in existence for over 30 years now, stakeholders engagement has mainly be through the conventional participatory approaches. WRUAs are expected to engage all stakeholders, solve conflicts, equal access to water resources etc. It is also paramount to mention that the membership to WRUAs is community-based (i.e. the executives are elected by the communities). And all game participants were members or belonged to respective WRUAs. The structural issues of the WRUAs and how gaming can 'change WRUA structural aspects' are beyond the objectives of this study, but I believe we can use game results to speculate. There are five interacting phases in the public debate on natural resource management: (A) agenda setting, (B) shared understanding, (C) commitment to goals, (D) means of implementation, and (E) re-evaluation based on monitoring. With only 'three pilot game sessions', this study focussed on the first two levels of stakeholders' engagement; (a) agenda setting, (b) shared understanding. Hence the focus on communication (subtractive dynamics e.g. a decrease in selfishness, and constructive dynamics e.g. a build up of collectiveness, knowledge gain etc), active participation, type of decisions made and their implications on water resources availability, etc. However, we can reflect the game results further to discuss and speculate on the other three levels (c) commitment to collective action, (d) means of implementation), and (e) re-evaluation. We have already alluded to this in* **Ln 465-468.** *It is also paramount to mention, that there is another paper (about to be submitted) with more ENGAGE gaming sessions, that has gone further to follow up the game participants 5-7 months later after playing the game. It captures changes in perspectives before and immediately after playing the game, and several months later.*

**I also have some specific comments:**

**Line 21 (Abstract):** What do the authors mean by "shared understanding", and understanding of what exactly? Could it be clarified?

*Response: Thank you for the comment. This refers to the second level of five interacting phases in the public debate on natural resource management. After level one, agenda setting (acknowledging there is a problem), the second level is "shared understanding" - understanding its causes and consequences. This has been mentioned in* **Ln 465**.

**Line 27 (abstract):** This sentence seems to point to the fact that the objective of the study was to assess the potential of the game to strengthen stakeholder engagement "next" (that is, after the game sessions were held). I do not think the paper demonstrates this. It demonstrates rather convincingly that there was active engagement during the game session but does not engage with whether or not this translated in strengthened engagement outside of the game session. I understand from the exchange with the community member who commented on the paper that this is not the point of this particular paper but then, maybe this sentence needs to be rephrased to indicate that what is being described in the dynamics of engagement during gaming session. Similarly, some clarity may be needed on line 125.

*Response: Thank you for the comment. This comment is related to the immediate comment above. In this study, we did not design to have comparisons e.g. comparing engagements with and without gaming sessions. Rather, we focussed on engagement patterns observed during gameplay given the*

*context described in the paper and in the above responses. In this paper we did not follow up the engagements outside the game environment, but three pilot game sessions provide important highlights that can help reflect on the potential of using serious gaming, given the context especially the use of conventional participatory approaches that have been in existence for over 30 years. We will revise the text to make it clear that the 'engagement' refers to the dynamics observed during the game.*

**Line 47.** The term "wicked problems" was first coined by Horst Rittel and Melvin Webber in a 1973 paper, which would be good to mention.

*Response: Thank you for the comment. This reference will be included in the revised version.*

**Line 54 to 56:** It is not clear how the 5 interacting phases of public debate had a bearing on the research conducted. Why do the authors refer to this work, and how useful is it to understand the process of designing and implementing game sessions? This is clarified around line 95. Consider slight restructuring on when this is mentioned so as to avoid repetitions?

*Response: Thank you for the comment. This is an important point and we will revise to avoid any repetitions. In the public debate on natural resource management, with or without the gaming, strengthening stakeholders' engagement is likely to follow the five interactive levels (A) agenda setting, (B) shared understanding, (C) commitment to goals, (D) means of implementation, and (E) re-evaluation based on monitoring. Gaming approach is coming in as an alternative to the conventional approaches, and testing the potential of serious gaming especially in the first two phases was a key focus of this study.*

**Line 58 and following:** I would not call IWRM an "approach". In my view it consists rather in a policy and discursive model than an approach per se.

*Response: Thank you for the comment. This is a good point of reflection. We will relook again on how we are using the term and revise.*

**Line 63:** can the authors briefly say what successes were actually achieved? In what terms was implementation successful and in which contexts?

*Response: Thank you for the comment. The successes of IWRM has mainly been reported on balancing the social, environmental, and economical issues of a basin or catchment. However, the power imbalances, inclusion, lack of common perspectives, collective actions, sustainable collaborations etc, remain a huge challenge. We will include the reported successes and their contexts in the revised version.*

**Line 107:** improved efficiency of what?

*Response: Thank you for the comment. This refers to improved 'efficiency of strategies' during stakeholder sessions (e.g. from low efficient strategy to a high efficient strategy i.e. revising individual/group strategies). This will be revised accordingly.*

**Line 101:** what does "this" relate to?

*Response: Thank you for the comment. I do not see 'this' in **Ln 101**, could you be referring to **Ln 112**? If this is the case, 'this' relates to 'studying communication patterns'…. The phrase will be revised accordingly.*

**Line 112 and following:** I am not sure what those sentences are meant to bring to the paper. They look as quite general statements in relation to games (what others have said on games), but it is not clear how much of this is actually being used in this particular study. Maybe they do not belong here

but should come earlier in the paragraph? I have a similar remark on communication: are the two sentences on that specific topic meant to discuss game "in general" or has this topic of communication during game played been a specific entry point of analysis of what has happened in the game session organized in this study?

*Response: Thank you for the comment. The sentences appear immediately after describing the importance of 'communication analysis' especially during engagement sessions. These sentences are important to reflect on in the introduction section because, during communication/engagements this could be in line with relational logic (value attached on how stakeholders relate to one another) or instrumental aspects (economic perspectives) or both. Furthermore, it would also be important to study communication patterns and emerging patterns by looking further into 'pressures' or 'external factors' influencing changes in strategies and decisions made by participants. For instance ' a participant revising strategies due to a government fine. This has been described in* **Ln 310, Ln 487, Ln 547-549.** *Yes, the communication analyses focussed on what happened during game sessions. We will relook at the sentences again to rephrase and highlight their importance.*

**Line 132:** what do the authors mean by "sentiment" and how did they assess it. What do they mean by "active" participation and how was it assessed.

*Response: Thank you for the comment. Sentiments refers to 'verbalized statements' expressed during gameplay. As stated in* **Ln 154**…. *The game sessions were video recorded to allow post-game analysis of sentiments. Each of the extracted sentiment/statement was subjected to a scoring scale described in* **Table 1**. *Active participation refers to 'enhanced communication among participants'-which was assessed by analysing the type and direction of sentiments e.g. directed to other participants, to the facilitator or as a spontaneous reaction to game outcomes.*

**Line 145 and following:** why does the discussion focuses on "substractive dynamics" only and not on "constructive dynamics"?

*Response: Thank you for the comment. It is not very clear what the referee means with only focussing on 'subtractive dynamics'.* **Ln 145** *is an extended description of 'counterfactual thinking' concept. For instance, conflicts and tensions would trigger thoughts about alternatives to these problems. If that is the case, then it is logical to assume that subtractive dynamics such as selfishness, tensions, conflicts would reduce with the build up of constructive dynamics such as knowledge gain, collectiveness, cooperation, collaboration etc. We have not only focussed on subtractive dynamics, if you look at* **figures 7** *and* **8**, *we describe and discuss both subtractive and constructive dynamics.*

**Line 160 and following:** how many WRUA are there in the sub basin/case study area? One for each sub-river?

*Response: Thank you for the comment. There are seven WRUAS in the selected study area. Yes, one WRUA represent one sub-river area.*

**Line 177 and following:** This points to my first major comment. How was this suite of problems identified and by whom? And more specifically, have the people who participated to the game sessions been involved in the definition of these problems. This needs to be clarified as it is largely from this that one can assess to which extent the process followed is fundamentally different from classic "participatory activities" implemented under IWRM processes.

*Response: Thank you for the comment. These comment has been addressed while responding to major comment no. 1. Yes, the catchment stakeholders were involved in definition of game components, refining of rules and processes that would increase interest in playing the ENGAGE game. However, it is important to point out that the actual participants involved in the three pilot game sessions were different from those involved in conceptualization of the game. As the referee mentions in this comment, this is one of the aspects why serious gaming approach differs from other*

*conventional workshops where the 'scientists' seem to take full control. In gaming concept, the 'scientists' play the role of 'facilitators' (from game conceptualization to actual implementation).*

---

## Author Response (AR3)

**Editor comments on the manuscript**

I read the revised manuscript in detail with great interest. I have still quite a few detailed comments, mainly of an editorial nature. I also have one more substantive comment, that was not raised by any of the two reviewers but which I still find relevant for the authors to consider. They may ignore it, or if they agree, may improve the paper further. I start with the substantive comment, which is followed by detailed comments of a less substantive nature.

*Response: Dear editor, thank you for taking time reading the manuscript. We have responded to each comment as indicated below:*

**1. Substantive comment**

The paper reports a rather successful application of a serious game, successful in the sense that it indeed appears that participants did learn as a result of the game, which is exemplified by some of the graphs. These show that in each of the three games that were played, there was a significant reduction of what the authors call "subtractive" dynamics and a significant increase of "constructive" dynamics (Figure 8). This I found impressive. However, this positive trend is associated with a general increase in profits that all individual players achieved (Figure 9), with one main exception, namely the livestock keepers in Teleswani sub-catchment, whose profit in the fourth run was slightly (but not significantly) lower than in the first run.

My inference: none of the players was significantly worse off at the end of the game. This was highly surprising to me. In particularly so, because the authors emphasised insistently from the start of the paper, that they were dealing with a wicked problem (used 13 times in the main text). As far as I understand, if there is a case of a wicked problem, there may not even be agreement what precisely the problem is, let alone its cause, and even less so its resolution. And if there is some kind of resolution, a "win-win" compromise is extremely unlikely - some, if not most or even all, may lose something.

This therefore raises a pertinent question: did the paper actually deal with a wicked problem? If it did, how then was it possible that none of the players was worse off at the end of each of the three games? Was there perhaps a flaw in the game set up that can explain this? Or didn't the paper deal with a wicked problem?

In my view we have to be careful (and economic) with labelling problems as "wicked" and only do so when we are really convinced they are. If we use it, let's be precise in defining the problem's wickedness. The paper defines it as "Problems, with many interdependent factors that make them very hard to solve, such as the differences in how humans view and interact with the dynamic physical environment" (lines 47-48). I am not an expert on this, so I am not in a position to assess how rigorous and adequate the used definition is.

Let's, for the argument, just assume that the problem central to the game was indeed wicked. Then there must have been a flaw in the game set-up. This flaw is not very difficult to identify. The set-up was such that the zero-sum game so typical for many water problems (and I guess in particular the wicked ones) could be circumvented, namely by creating new water. And not a little: at least 50% additional water was created in each of the three games sessions (line 594). This implies that the game "unwickeded" the problem! And if that's indeed the case then the frequent references to wicked problems in the paper may actually be a problem.

*Response: Thank you for this substantial comment. To begin with the motivation for this research, the research emanated from the complex human-water challenges in the case study area. that can be described as 'wicked' in our opinion (i.e. many different actors, with different needs, different perspectives, which makes identifying the problem and the solution difficult). We presented the context of the case study area, where there have been human-water challenges including conflicts for over many years. Even the establishment of river basin organizations such as WRUAs (who have been in existence for over 30 years), have not yielded sustainable results. The situation has deteriorated (even with reported annual fatal conflicts), which can be linked to changing hydrological dynamics and evolving human dynamics e.g population growth. Our argument is that WRUAs have been engaging communities through conventional approaches e.g. workshops, community discussions, group discussions etc.*

*In the scientific arena, among a range of participatory approaches, the development and use of serious games has gained prominence as a tool to stimulate discussion and reflection among stakeholders about sustainable resource use and collective action. Hence, in **Ln 245**, we write ...."developing and testing an alternative participatory approach such as a serious game was considered timely"......*

*We agree with the comment, however the serious games can be viewed to make the problems 'less wicked' by simplifying the real-world complexity, highlighting common pull resource elements, make the situation more*

insightful and possible improvements. The participants in a game setup are not likely to forget the others or the future, it is one way to get out of the 'wicked trap'. In **Ln 158**, we write….."The purpose of this study was to assess the potential role of the ENGAGE game in strengthening stakeholder engagement toward addressing complex human-water challenges of a river catchment"…..

By design, there are two phases in the actual game play. Phase one (which was meant reveal the 'wickedness') mimics reality, whereby individual values and preferences of the players were allowed to shape the game results. Phase one was meant to prepare players for phase two. By unpacking the wickedness in phase one, players were sensitized to the different needs of the other players and triggered to jointly look for a more sustainable alternative ways of management. In the second phase (i.e. a final round or 'reflection' round), the players were guided to reflect on the game results and experiences in phase one and think objectively about what the potential solutions to the human-water challenges could be (observed in phase one). Therefore, it is not that there is a flaw in the game (that we are aware of), but the actual game set-up, can explain the reason behind your inference that the game "unwicked" the human-water problem. Undoubtedly, a different game design and set-up would have yielded different results. In the discussion section **Ln 623**, we write…. "conducting several game rounds without subjecting the participants to a final reflection round may be an alternative way to assess the emergence of gaming outcomes"…. This would have resulted in game outcomes that would potentially reveal worse case scenarios even in the final round. However, we argue that this would require a larger number of game sessions to gather a sufficiently large sample size of game sessions in order to draw meaningful patterns….

**Key changes made in the manuscript:**

As a reminder to the reader and reduce the perception of a flaw in the game results, we have revised the following text as part of the limitation of the study (reminding the reader of the two phases of the gameplay):

………"We argue that conducting several game rounds without subjecting the participants to a final reflection round may be an alternative way to assess the emergence of gaming outcomes. Without a reflection round, we speculate some game sessions would have ended with worst case scenarios, which still would have been important results as an emphasis of 'wicked' nature of human-water challenges. Although this would require a larger number of game sessions to gather a sufficiently large sample size of game sessions. Assessing game outcomes from such a larger sample size may be desirable to study more patterns and more emergent patterns of this gaming approach"…...

**2.   I have one additional query, which I address below, namely how the water harvesting was modelled.**

So a criticism against this paper could easily be made as follows: the game set up was designed for success. This obviously compromises the realistic nature of the game, and its applicability for real world problems. In this light, I would have found it appropriate if the authors had played a counterfactual game without the option to make new water available (through water harvesting), and assess whether similar learning patterns did emerge or not.

I am not arguing that the authors should do this. I want to authors to consider this criticism and if needed make some more elaborate and grounded arguments in the discussion. Currently in the discussion it is rather superficially argued and weak, namely: "… guiding stakeholders to practically understand the importance of rainwater harvesting and storage is an important factor for water availability in the river catchment. This finding concurs with other studies that opine that rainwater harvesting can be a feasible solution for water scarcity problems during dry seasons (Velasco-Muñoz et al., 2019; Irshad et al., 2007)." (lines 598-602)

To argue for supply-side solutions is what is the conventional solution to most water problems, as if there would be no limit to "squeezing the cow" (Waalewijn, 2002), and as if that would really resolve the problem. Refer in this context also to the "reservoir effect" postulated by Di Baldassarre et al. (2018). (The references that the authors give in the above quote are not very convincing to me.)

Finally, and to close this substantive point: it is important for the serious reader to know how the water harvesting was included/incorporated in the game set up. Normally, the more water you harvest, the more effort you have to put in and the lower the yield will be. So there are clearly and significant decreasing returns to investment for this. Was this taken into account? I understand from Supplement 2, that the ability of players to harvest additional water was "randomized". I don't know what exactly this means, and why this was chosen to be done. What real world situation is this supposed to mimic?

**Response:** Thank you for the comment. Regarding the suggestion for "a counterfactual game without the option to make new water available (through water harvesting)", we think this is readily addressed by the development

of solution space. The solution space (with 1000 possible outcomes) gives the realm of possibilities based on participant choices in the ENGAGE game. Fig 11 shows the actual game results with rainwater harvesting and if no rainwater harvesting was implemented. Based on the solution space in Fig 11, it is possible to deduce that there are possibilities of some game outcomes (water availability results) falling below the critical threshold. However, the actual game results revealed increasing water availability, with harvesting decisions.

Hence, given the purpose (…."to assess the potential role of the ENGAGE game in strengthening stakeholder engagement toward addressing complex human-water challenges"……), our argument in the quoted Ln 598 is that the ENGAGE game …."guided stakeholders to practically understand the importance of rainwater harvesting"…..

Regarding the question on…."how the water harvesting was included/incorporated in the game set up"…. Yes, the more you harvest, the more the investment costs, hence lowering potential net income… in the supplement 1, the rainwater harvesting costs is set to be high at KES 50,000, equivalent to the costs of opening a new agricultural patch. Hence, the returns to investment were considered.

I do not understand the comment on …."the ability of players to harvest additional was randomized"…. However, in my interpretation the comment refers to randomization described in Table S1. If this is the case, we did not randomize the ability of the players to harvest water. The randomization described in Table S1 was in the construction of the solution space where randomization between minimum and maximum values (of different game elements) was done to ensure a complete representation of all possibilities (in the 1000 runs).

**Changes made to the manuscript:**

We have revised the quoted text in the discussion section to also include 'anticipated' effects of the water harvesting solutions as follows:

……" There was a decline in water availability in the initial game rounds and then an increase toward the final game rounds. This could be attributed to rainwater harvesting decisions made by game participants in the final game rounds which saw water availability increasing between 50% and 91% in the three game sessions. Therefore, despite increases in agricultural land expansion and livestock units, guiding stakeholders to practically understand the importance of rainwater harvesting and storage is an important factor for water availability in the river catchment. This finding concurs with other studies that opine that rainwater harvesting can be a feasible solution for water scarcity problems during dry seasons (Irshad et al., 2007; Velasco-Muñoz et al., 2019). However, these are supply-side solutions which can readily fit the context of water scarcity problems where there is urgent need to provide 'additional water' to quell existing conflicts as conceptualized in this study. Overall, there could be thresholds to the extent to which the supply-side solutions are sustainable. Increasing water storage capacity, despite improving water supply can increase stakeholders' vulnerability due to reservoir effect (Di Baldassarre et al., 2018), human displacement (Asmal, 2000; Kuil et al., 2018), inefficiency due to excess water 'reallocation' (Kuil et al., 2018), among other anticipated effects."…….

**3. Detailed comments**

1. Line 1: I find the term "human-water-related challenges" confusing and ambiguous (it occurs 7 times in the text, and it features in the title of the paper). Omitting "-related" would instead make it straightforward, and this is also used (also 7x) in the text.

**Response:** Thank you for the suggestion. We have removed "-related" from the text.

2. Line 29: The addition "explored within a game environment" I found not very helpful.

**Response:** Thank you for the suggestion. This has been removed

3. Line 69: "Firstly, …" This is, as far as I could see, not followed up by "Secondly, …".

**Response:** Thank you for this comment. We have revised the text as follows:

……..Firstly, IWRM does not directly account for the dynamics of the interactions and feedback between water and people (Sivapalan et al., 2012). Secondly and most importantly, IWRM typically adopts participatory methodologies such as workshops, focus group discussions, dialogue groups, etc……..

4. Line 78: Not clear whether (UNSDG, 2020) is used here as a standard. Unclear

**Response:** Thank you for the comment, the intention was to give the reader a picture of existing stakeholder engagement standards at local or country level. We have revised the text as follows:

……..Even stakeholder engagement standards such as AA100AP (Kim et al., 2018) applicable at local level, or (UNSDG, 2020) applicable at national level, among other standards, fail to create a learning space that goes beyond participation and allows stakeholders to directly engage with………

5. Line 80: Although we see this with increasing frequency, it is in my view not good practice to use "(Bielsa and Cazcarro, 2015) underlined the need …" This should be: "Bielsa and Cazcarro (2015) underlined the need …" See also Lines 226, 519, 640

**Response:** Thank you for the comment. This is well noted. This mostly happens when using referencing softwares such as Mendeley. We have checked and manually revised the references accordingly.

6. Line 83: "This will help …" Always? Unlikely. So I would prefer: "This may help …"

**Response:** Thank you for the comment. We have revised accordingly.

7. Line 86: Why should Speelman et al. (2021) have his initials included in the reference?

**Response:** Thank you for this observation. The initials have been removed.

8. Lines 92-124: This is largely a new text and is inserted in a paragraph that argues that serious gaming is an alternative approach for participation. In these new sections there are also frequent remarks concerning participatory modelling, but it remains not very clear why this is included in the argument. Especially the following sentence for me created some confusion:

"There are different ways to increase engagement of participants during workshops, such as participatory mapping, experimentation with art-based visuals, etc, however, these cannot be viewed as collaborative modelling. Basco-Carrera et al. (2017), attempts to differentiate what can be considered as 'participatory modelling' and 'collaborative modelling'." (lines 98-101)

The first part of this quote suffices (until the word "etc"), whereas the remainder confuses and can better be omitted. The paragraph that follows starts with serious gaming, but the second half of the paragraph again deals again with companion modelling. It is then, rather cryptically concluded that "As aforementioned, the politics that shape conventional processes (e.g. the influence of the 'outsider') are dealt with in the gaming approach through an iterative process that evolves with participatory modelling …" (lines 120-122)

I rally fail to understand what this sentence wishes to convey. And it remains obscure to me how (and why) serious games and participatory modelling appear to be intimately linked.

**Response:** Thank you for the comment. In our argument, the process of developing a serious game is done through a participatory modelling (a purposive learning process that incorporates stakeholders views in developing the game), and this is how we link the two concepts. The process of developing a serious game is similar to that of participatory modelling, as the first steps are the same with one leading to a computer model and the other with a game, but both are models (simplified representations of reality).

Despite the game development, the actual playing of the game can also be described as participatory modelling. This is because, the serious gaming is purposive (i.e. "serious") for instance, games used for social leaning where stakeholders are engaged to explore possibilities through participatory modelling that may give rise to different results using the same board game elements. Therefore, there is a close link between serious gaming and participatory modelling.

Regarding companion modelling, this part was inserted as a response to a reviewers comments regarding 'outsiders' influencing the game development. However, the 'outsiders' influence is diffused through a co-construction approach (where the designers and the participants collaborate to define game development) commonly referred to as companion modelling approach (but can easily be termed as participatory modelling approach).

**Changes made to the manuscript:**
To minimize the confusion to the reader, we have removed this text in Ln 100:
-
- *To align with the general text in the manuscript, we have also replaced 'companion modelling' with 'participatory modelling'*

9. Lines 116-117: "However, the companion modelling approach needs to be improved to clearly define the horizontal and vertical dialogues by involving all stakeholder at all levels (Barnaud et al., 2008)." Why is this important? Can't this sentence be omitted?

**Response:** Thank you for the comment. This part has been removed.

10. Line 118: "stallholders" -> stakeholders

**Response:** Thank you for the observation. This has been corrected

11. Line 211: "The aridity values in the catchment change drastically between …" -> "The aridity in the catchment changes drastically between … "

**Response:** Thank you for the suggestion. This has been revised accordingly

12. Line 243-244: "… developing and testing an alternative participatory approach such as a serious gaming is timely." Rephrase into "… developing and testing an alternative participatory approach such as a serious game was considered timely."

**Response:** Thank you for the suggestion. This has been revised accordingly

13. Line 251: "the ARDI approach" Explain what ARDI stands for.

**Response:** Thank you for the comment. This has been inserted in the text.

14. Lines 253-254: the three subcatchments have already been named earlier. No need to repeat.

**Response:** thank you for the suggestion. This has been revised accordingly.

15. Line 256: "… in three sub-catchments." -> "… in the three sub-catchments."

**Response:** thank you for the suggestion. This has been revised accordingly.

16. Line 262: I would omit this entire line "Name of the Game: ENGAGE_v1 – "Exploring New Gaming Approach to Guide and Enlighten", and insert in the next line an explanation of what ENGAGE means (i.e. "Exploring New Gaming Approach to Guide and Enlighten")

**Response:** thank you for the suggestion. The line has been removed

17. Line 291: "… agricultural activities … " This may create some confusion, because this excludes livestock, which follows later. Then this is about "arable agricultural activities".

**Response:** thank you for the suggestion. This has been revised accordingly.

18. Line 296: add "area" to grazing to make "grazing area"

**Response:** thank you for the suggestion. This has been revised accordingly.

19. Line 303: insert a missing parenthesis.

**Response:** thank you for your observation. This has been added

20. Line 309-310: "The effects of changes in the water balance are however heavily felt in the downstream zone." I would prefer: "The effects of changes in the water balance are however most heavily felt in the downstream zone."

**Response:** thank you for the suggestion. This has been revised accordingly.

21. Lines 375-378: "Water availability was accounted as the difference between the water generated from the water tower and total water demand. Rainwater harvesting was considered as 'additional water' for the board game, as this was done during the transition of game rounds." This may be correct, but this may create some confusion when interpreting the graphs (the Figures in the next section), which I have tried to indicate (see below).

**Response:** Thank you for the comment. We have responded to the specific comments below.

22. Line 388-389: "The number of livestock that survived within the board game system (at the end of each game round) was observed to be fewer than the available stock at the start of the game." Check whether this is correct or should state "… to be equal to or fewer than …"

**Response:** Thank you for the comment. This has been checked and revised accordingly.

23. Figure 4:- Somehow Include somewhere (perhaps in a new series of graphs) the water availability per run (this should include and distinguish the "normal" water availability (which varies per run due to the dice thrown), as well as the water generated due to water harvesting initiatives of farmers during the run).

**Response:** Thank you for the comment. The water availability per run is already provided in Fig 10.

23.1. For each figure: Use identical y-axis throughout, so that the graphs are easily readable and comparable.

**Response:** Thank you for the comment. We have revised the Y-axis to ensure equal units.

23.2. Check why Figure 4G has four runs/rounds (shouldn't this be 3?)

**Response:** Thank you for this observation. We have checked and revised Fig 4G.

23.3. Explain in the caption of this figure the meaning of R1 to R4

**Response:** Thank you for the comment. We have added the explanations for R1-R4.

24. Figure 5: Identical Y-axis, please

**Response:** Thank you for the comment. This has been revised.

25. Figure 6: Same: identical axis

**Response:** Thank you for the comment. This has been revised.

26. Line 425: "Subtractive dynamics seem to reduce towards zero in the succeeding game rounds …". Consider to rephrase this as: "Subtractive dynamics seem to reduce in the successive game rounds …"

**Response:** Thank you for the comment. This has been revised.

27. Line 427: "… had a reverse trend … " consider to rephrase this as "… had a reversal …"

**Response:** Thank you for the comment. This has been revised.

28. Line 429: "... with the increase in knowledge gain and use of plural pronouns …" -> "... with the increase in knowledge gain and the increased use of plural pronouns …"

**Response:** Thank you for the comment. This has been revised.

29. Line 431-432: "… knowledge gain maintained to continuously increase throughout the different game rounds" -> "… knowledge gain continued to increase throughout the different game rounds"

**Response:** Thank you for the comment. This has been revised.

30. Figure 7: Consider reducing the values of the scores in all graphs of this figure to only one digit

**Response:** Thank you for the comment. This has been revised.

30.1. For Figs 7B, 7D and 7F: the colours orange and red are difficult to distinguish. Why not use blue for the red.

**Response:** Thank you for the suggestion. This has been revised.

31. Line 454: "… made by game participants (Fig SThe results show that …" -> "… made by game participants (Fig S2). The results show that …"

**Response:** Thank you for the comment. This has been revised.

32. Lines 460-462: "Plotting the game session results in the game solution space, showed that investment in agricultural expansion in the midstream zone may not necessarily lead to an increase in net profit, based on the best-fit line in Fig 9B." I didn't understand this. Perhaps I did not understand what the "best-fit line" actually meant.

**Response:** Thank you for the comment. The line of best fit is used here as an output of regression analysis (predicting the relationship between the net profit and the number of agricultural practices). To minimize confusion, we have deleted this part from the text…. ""…

33. Figure 9: It is critical to use identical Y-axes here, as this would more clearly show that whereas most profits increased, those of the irrigators increased much more than that of livestock keepers.

Response: Thank you for the comment. This has been revised

33.1. Is the net profit per player or per group of players? What was the inequality index (or the variation of the profits between the individual players in each group)? Was it similar for the three groups (up, mid and downstream), or different? Wouldn't that give important additional information? Why was inequality between players not included as an indicator?

**Response:** Thank you for the comments and questions. The net profit in Fig 9 is calculated for each game round (based on the profit results of all players). This study did not analyze the inequality index between the individual players. The focus was more into observing the system level outcomes (i.e. catchment level perspectives). An analysis focusing on household level would best fit the suggestion of developing of individual economic indicators/indexes and especially where there is a focus is on economic dynamics.

33.2. What does the best fit line show?

**Response:** The line of best fit is used here as an output of regression analysis (predicting the relationship between the net profit and the investment in agricultural patches or livestock numbers).

34. Line 467-468: "However, the actual game results showed a reverse of this trend, where water availability increased especially toward the final game rounds (Fig 10)." I cannot see this in Figure 10: it reduced for Nyuki, it increased only very slightly but not significantly for Teleswani, and it did increase for Timau. The best fit line showed that the water availability decreased with the expansion but I still don't understand what this means.

**Response:** Thank you for the comment. The line of best fit is developed using the solution space simulation (the grey points). The actual game results were plotted inside the solution space. Using the line of best

fit/regression prediction, the water availability decreases with increase in the number of agricultural patches. However, the actual game results had a different trends where an increase in water availability was observed. This was linked to water harvesting decisions (see Ln 476:…." *The results reveal that an increase in water availability toward the final game rounds seems to coincide with increasing water harvesting decisions made by game participants (Fig 11A)"*…..

**Changes in the manuscript:**

To minimize the confusion, we have revised the text to indicate the specific game sessions where the increase was observed as follows:

….."*However, the actual game results showed a reverse of this trend, where water availability increased in Teleswani and Timau game sessions especially toward the final game rounds (Fig 10)"*…..

35. Figure 10: Does the water availability in this graph include the additional water created by water harvesting?

**Response:** Thank you for the question. Yes, the water availability also accounts for additional water.

35.1.Isn't the second line of the caption wishful thinking? Why is this remark included in the caption of this figure? If it is important, shouldn't it feature in the main text. Does it?

**Response:** Thank you for the comment. This part has been removed from the caption and placed in the main text.

36. Figure 11: For readability and comparability of both graphs it would be better to use the same colours for the lines of the three sub-catchments. The difference in Fig11B between with water harvesting and without could be indicated by e.g. using broken lines for the latter.

**Response:** Thank you for the suggestion. This has been revised accordingly.

37. Line 501-502: "Plotting of actual game results on net profits in the solution space revealed that upstream and midstream players aimed for profit maximization, compared to the pastoralist players in the downstream zone." Is this correctly phrased? Perhaps the livestock farmers also aimed to maximise profit, but simply could not achieve it?

**Response:** Thank you for the comment. The sentence has been revised as follows:

…."*Plotting of actual game results on net profits in the solution space revealed maximum profits for the upstream and midstream players, compared to the pastoralist players in the downstream zone"*…..

38. Line 515-516: "For instance, the need to adopt rainwater harvesting as a strategy to reverse the negative consequences spontaneously emerged among the players." This remark came as a surprise to me and to me is highly suggestive. This is because it was earlier indicated in the paper that this was part of the game's design (see e.g. line 307: "The game participants may react by investing in rainwater harvesting …"). So how "spontaneous" was this?

**Response:** Thank you for the comment. Yes, rainwater harvesting was one of the possibilities in the game design that could have been adopted by players. However, in the actual game play, this was not 'forced' among players and the need for additional water was 'spontaneous' in the successive rounds. In reality, rainwater harvesting is not a common strategy at household level. But when stakeholders are brought together in this game sessions, they need for harvesting additional water as a strategy to the water scarcity manifests. Actually, based on Fig 4, rainwater harvesting was absent in the 1st round, but increased in the subsequent game rounds.

39. Line 526 and 531: "emigration" -> "migration". (Emigration has the suggestion of permanence, whereas migration has a more generic and open meaning, which I think is more appropriate for this context.)

**Response:** Thank you for the comment. This has been revised.

40. Line 568-569: "The next steps on commitment to goals and means of implementation would depend on the way the game is part of a longer-term process of interactions." I would add the verb "made", as follows: "The next steps on commitment to goals and means of implementation would depend on the way the game is made part of a longer-term process of interactions."

**Response:** Thank you for the suggestion. This has been revised.

41. Line 676: "Water resources management stakeholders can work with the ENGAGE game as a starting point." This last sentence of the paper I didn't find very convincing. This may be true for the few irrigators and livestock keepers in the Ewaso Ng'iro, but the paper hasn't demonstrated that it can be used elsewhere.

**Response:** Thank you for the comment. We have revised the text by adding Ewas Ng'iro context as follows:

......."Water resources management stakeholders can work with the ENGAGE game as a starting point for catchments with similar context as Upper Ewaso Ng'iro basin".

42. References list: The references contain some odd entries, see lines 844, 850, 962, 1033, which need to be corrected according to the journal's standards, and then of course the references to these entries in the main text need to be corrected accordingly

**Response:** Thank you for this observation. This has been checked and corrected.

43. Incomplete references: lines 816, 856

Response: Thank you for the comment. The references have been corrected.

**Reviewer 1**: **Wim Douven,**
Dear Charles and team, I enjoyed reading your manuscript. I find the topic very interesting, and it is good to have another study on serious games. Below some comments, questions, that might be helpful in finalising the manuscript. I am happy to hear you enjoyed reading the manuscript. Thank you for the positive remarks on the manuscript and constructive suggestions and comments.

**Comment 1:** In the section on IWRM (line 66 and further), you mention that workshops do not promote different experts to engage with wicked problems. But doesn't that depend on how you design such workshops, like how you design serious games ..? Thank you for the question.

**Response:** Yes, we agree that that there are many different participatory methods developed especially to promote the engagement of 'local experts'. One of the most familiar is focus groups, but many others are around including *hands-on' interactive activities e.g. participatory mapping, experimentation with art-based visuals, etc*. Designing serious game sessions may differ from designing workshop sessions. Firstly, serious games are co-designed where both the 'scientists' and 'local experts' are involved in the development of serious games (way before actual game sessions). In the workshop sessions, the scientists' are the most active persons, e.g. when making PowerPoint presentations, explaining an idea, reporting case studies, or simulating a model and its possible results. The participation of 'local experts' is relatively 'passive' in the workshop sessions e.g. sitting and listening to presentations made by scientists. Some workshops may divide participants into small groups for discussions, experimenting, and demonstrating workshop issues. But in most cases, the group discussions tend to have active and less active persons - *in **Ln 659** of the manuscript, we have argued this could be due to power differentials among participants*… Workshops often are not the right way to deal with wicked problems.

There is a difference in the game sessions. Serious games provide a 'buffer'  to mitigate powerplay and hierarchy, make it more easy for participants to express more freely, since it is 'just' a game. Serious games game provide a boundary object commonly not found in workshops. The 'scientists' are less active, and tend to play the passive role of facilitating the process (especially at the start of the game rounds), while participants are the most active persons (in small groups) right from the start to the end of the sessions. The facilitation role tends to decrease as the game progresses, and at some point 'scientists' become the 'observers' of what is happening in the different small groups of gameplay. Scientists tend to learn more this way, as opposed to when they are the main presenters in conventional workshops. The game participants are vigorous as they interact directly with 'virtual systems' such as Board games and also actively engage one another (e.g. disagreeing, setting rules, seeking alternative strategies, etc), the scientists do not control the actual gameplay. *In **Ln 654** of the manuscript, we have argued that '…..since each participant in the game has a role to play, they have a stake during the game. They tend to be active in pursuing their stake while focusing on their roles…'*

We agree that if we design workshops similarly to game processes (including having game-like engagements during small group discussions), then designs of workshops and game sessions could both promote different experts to engage in a meaningful way. Still, in most conventional workshops the 'scientists' are the main 'active persons' while 'local experts' are the 'passive' participants. By design game sessions reverse these roles.

*Revisions made in the manuscript:*

We have revised manuscript to expound our argument on conventional workshops in the (see Ln 93-103). In our revisions, we have compared conventional approaches and modelling to serious games (especially on the role of 'outsiders' dictating the processes), black-boxed versus open processes in games, included reported limitations in serious gaming.

**Comment 2:** I note that the lit. review on games is overall very positive on its effects on finding joint solutions etc.., are there no limitations and challenges reported in literature ?  You mention one in your discussion in line 520. Did you come across other limitations, challenges in the sessions? Thank you for the questions.

**Response:** This study focussed on the literature that has attempted to report on the 'effectiveness of gaming approach'. Hence the purpose of this study was to 'investigate these opinions/claims' by assessing the potential role of serious gaming in strengthening stakeholder engagement. This study contributes to the debate on the 'effectiveness of serious gaming' including how some results contradict past studies see **Ln 578**.

*Revisions made in the manuscript:*

The following text has been added in the revised version (see Ln 103-125):

*…… In their study, Flood et al. (2018) conducted a review of 43 serious gaming publications and identified the major shortcomings to effective game design and engagement as; one-off engagement (i.e. several game sessions are needed to enhance learning), capturing complexity without overwhelming the stakeholders, future planning (i.e. linking game results to plan an uncertain future). Serious games are also limited on the number of stakeholders who can be involved in a single game session, a constraint that raises the politics of who should attend the game(s) and why? (Wesselow and Stoll-Kleemann, 2018; Edmunds and Wollenberg, 2001). Studies have also reported that social differentiations and power asymmetries have greater influence on the outcomes of a participatory process (Barnaud and Van Paassen, 2013; Mathevet et al., 2014). Both the facilitators and the stakeholders have various degrees in which they can influence the participatory process (Jonsson et al., 2007). Serious gaming can also exacerbate the contests of power due to constraints of simplifying the complex real worlds, balancing the interests of the locals and the 'outsiders', and different perspectives of the present and future (Venot et al., 2022). A co-construction process also referred to as companion modelling approach where the designers and the participants collaborate to define the entire process is seen as a way to improve legitimacy of the participatory process and enhancing multi-stakeholder cooperation (Barnaud and Van Paassen, 2013; Barreteau et al., 2014; Basco-Carrera et al., 2018; Étienne, 2014). However, the companion modelling approach needs to be improved to clearly define the horizontal and vertical dialogues by involving all stakeholder at all levels (Barnaud et al., 2008). In general, the quality of participatory process depends on how biases and interests of all stallholders, including 'outsiders' are balanced (Biggs et al., 2021; Daniell et al., 2010). As aforementioned, the politics that shape conventional processes (e.g. the influence of the 'outsider') are dealt with in the gaming approach through an iterative process that evolves with participatory modelling (Marini et al., 2018a; Speelman, 2014a; Speelman et al., 2019b; Rodela et al., 2019b; Barreteau et al., 2014). Hence, this study can be viewed to have done something different from the conventional participatory approaches (such as workshops, where 'outsiders' dictate the process) by creating a different type of collaborative engagement and a 'safe environment' for stakeholders.……*

**Comment 3:** In game conceptualisation (Section 2.2), it might help the reader to give a brief summary about sessions, rounds, and phases, and how they relate to the catchment / sub-catchments. Thank you for the suggestion.

**Response:** During this stage of game conceptualization, there was no playing the games per se, but this is the 'setting stage', which involved bringing all possible ideas to the table, to help in crafting a serious game that fits the context of the case study area. The ideas were gathered from past studies, through the ARDI approach, and conducting community discussions with communities in three sub-catchments. Game testing sessions are described in sub-section 2.3, where feedback was key to help in refining the final game.

***Revisions made in the manuscript:***

To guide the reader on understanding this stage of game conceptualization, we have revised the manuscript by adding the opening sentence as follows (see Ln 248-249):
…..This stage involved gathering all possible ideas, to help in drafting a serious game that mimics the context of the case study area.….

**Comment 4:** Data collection (Section 2.4) is there after the step 'modeling the game solution space' not also a step where you collect data about discussion and feed-back, which might result in adjustments of positions and solutions .? As it is an interactive process, it is collecting data and providing data, right? (so feed-back, and giving space for learning)? Thank you for the questions.

**Response**: The data collection description in sub-section 2.4 focussed on data collection based on the final ENGAGE game_v1 described in Box 1 and Supplement 1. To ensure comparability of data across different game sessions, no further adjustment was made and v1 of the game was maintained (as developed) in the three sub-catchments. Yes, there is a feedback session at the end of playing the games, where participants are allowed to give their feedback and key lessons on the game sessions. The qualitative feedback was useful in crafting some of the reflections included in the discussion section e.g. in **Ln 638** … the *game environment allows for real-time reflection through the creation of a fictional setting and a common pool for the stakeholders to explore decisions and impacts simultaneously'*……in **Ln 654** …….*'each participant in the game has a role to play, they have a stake during the game. They tend to be active in pursuing their stake while focusing on their roles'*…….

***Revisions made in the manuscript:***

We have revised the manuscript by adding the following sentence (see Ln 357-360):
*….At the end of each game sessions, post-game feedback sessions were also conducted where participants were allowed to give their feedback and key lessons on the game sessions. This qualitative feedback was useful in understanding participants perceptions and reflections, which were critical in qualitatively discussing the game results of this study….*

**Comment 5:** An important discussion point you touch upon is how the game results translate to real life situation (line 560 and further). I wondered was this discussed in the sessions, at the end? How the game was valued, also for real-life? That could already give some initial insights? Thank you for the questions.

**Response:** Yes, a conversation on translating game experience to real-life situations was a key discussion point at the end of the game sessions. There was high positivity on the usefulness of the gaming approach in promoting sustainable behaviors, and all game participants indicated they would adopt a few lessons from the game. However, this manuscript recommends further follow-ups to assess the impact of serious games on strengthening stakeholders' engagement and maintaining sustainable behaviors in real life.

***Revisions made in the manuscript:***

We have revised the manuscript as indicated below (see Ln 672-675):

*….Whether the results realized in a gaming environment could impact reality is still unclear. Post-game discussion revealed positivity on the usefulness of the gaming approach in promoting sustainable behaviors, where game participants indicated willingness to adopt a few lessons from the game. However, it may require more game sessions and long-term studies to assess the impact of serious games on strengthening stakeholders' engagement and maintaining sustainable behaviors in real life….*

**Reviewer 2: Comment by Jean-Philippe Venot**

This is a paper that investigates the use of a serious game to discuss issues of water management at watershed level in Kenya. I enjoyed reading the paper; a lot of work has gone into the design, implementation and analyze of the research. I think this type of paper reflecting on the use and scope of such participatory approaches for improving water governance is important.

That being said, I feel the paper shares what I see as a bias in the existing literature on games for sustainable natural resources management, that is, a rather positive outlook on the changes/transformations these games may trigger and a focus on their promises and potentials – that is only partly supported by what is actually described in the papers (that is what happens during "game sessions" as opposed to "in the real world").

*Response: Dear referee, thank you for the acknowledgment of the relevance of our research exploring the potential of participatory approaches such as serious games for improving water resources management and governance. We plan to address your major five comments and specific comments as shown below:*

(1) **MAJOR COMMENT 1: Better clarify how the process they steared was fundamentally different (or not) from other more classic participatory approaches notably in relation to co-production of the tool (as opposed to a contrasting active game with passive attendance to workshop).**

Line 66 and following: The authors attribute some of the shortcomings of IWRM to the fact that IWRM projects and policies make use of "classic" participatory approaches that would not really allow different actors to engage with the "wicked problem" at hand. In other words, participation in IWRM is not real participation. This is a well-known critique of IWRM, which I share in part, but I think there is a danger of building a strawman for the wrong reasons here, just to make the point that games do not share these shortcomings. I agree when the authors answer the previous community reviewer that games allow a different, more active, mode of engagement than classic workshop based that use power point presentation. But there are many ways to conduct workshop that can provide meaningful engagement of participants: participatory mapping, experimentation with art-based visual, etc. are some of these options.

I think the issue is less about the modalities of facilitation of workshops (and the tools used) than how multiplicity of knowledge is handled and how/who decides on the terms of the debate. Notably, who ends up deciding what the 'wicked problem' is. Games can actually reify the 'wicked problem' on the basis of an outsider/expert view; if so I would argue that the nature of engagement of participants remains as limited as in other approach or at least is constrained in major ways.

What bothers me is that the phrasing of this paragraph (and many others) seems to indicate that the authors hold the view that there is ONE "wicked problem" (likely a lack of coordination between downstream and upstream users). It makes it seem as if this problem existed almost independently of the people who participate, being identified à priori by a subset of actors (though there is scope for refining the problem at hand through the use of the game itself). If that is the case, then, the game developed is likely to have very similar shortcomings than "classic" IWRM based approach.

I think the critique of IWRM approaches would be more powerful and convincing if it were to be done at the level of how are the terms of the debate/participation set and by whom (as opposed to looking at the specifics of interactions and their presumed passivity), and then to show, how this particular study did something different, hence created the room for another type of engagement by the people who participated in the game session. The point here is to pay a bit more attention to the politics that framed the entire research process and operated silently in the background as opposed to take the dynamism of the game sessions as a sign of pro-active engagement and changes to come in the watershed. The authors may want to look at https://hess.copernicus.org/preprints/hess-2023-164/ currently under review in the same journal. This paper reflects on some of those issues around the practices of hydrological modeling not games, but I think there are many similariries.

I do not doubt the authors have done something quite different from "classic passive workshops"; my qualm is with the way authors describe the differences. Instead of making clear what the differences were, the way it is written points to fundamental similarities in the assumptions underlying the approach (which may not have been there). I feel clarifying this point through some reformulation is important and would strengthen the argument of the paper.

*Response: Thank you for the detailed comments. We agree with the referee that the lack of 'real participation' is a well-known critique on IWRM. We are happy to see the referee agreeing with our response to the community*

*comment that serious gaming is a different, more active, mode of engagement than classic workshops. Interestingly, the reviewer also confirms that there are many ways of conducting workshops that can provide meaningful engagement of participants (particularly the inclusion of 'hands-on' interactive activities in the workshops e.g. participatory mapping, experimentation with art-based visuals, etc). This re-affirms our arguments on the need for 'hands-on interactive activities' in the conventional participatory approaches. And not just the inclusion of 'hands-on interactive activities', but those activities should yield results that can inform or indicate how people interact with their environment and opportunities for collectiveness and collaboration. The fact that there are no uniformity/standard ways of conducting participatory engagements in IWRM demonstrates the need to develop and evaluate different participatory approaches. Serious gaming is one of the possible strategies to be investigated more deeply given the growing complexity of human-environmental issues.*

*Given the context of the case study area, where there have been water-related conflicts annually (even fatal conflicts), and the fact that under normal circumstances, the downstream communities do not sit or see eye to eye with the upstream communities **(see Ln 534)**; developing and testing an alternative participatory approach such as a serious game is timely. This is especially so since, the River Basin Organisations (i.e. WRUAs) in the case study area have been in existence for over 30 years now, and mostly the communities have been exposed to the conventional engagement approaches e.g. workshops, community discussions, group discussions etc **(See Ln 230).** And yet the situations are getting worse each year, with the national government 'reacting to calm the conflicts' but with no permanent solutions **(See Ln 239)**.*

*In conventional engagement approaches for IWRM, the leading scientists have developed hydrological models that are presented to the stakeholders/participants to shape human-water discussions. Different scenarios can be tested and results presented during a workshop to allow deliberations on the way forward towards resilience. This in some way has a sense of 'top-down strategy'. In a serious gaming process, all stakeholders (scientists/non-scientists/ including low-educated) define the problems, priorities, and test scenarios themselves, and 'revise' their perceptions based on experiences from playing the game with less 'scientists' input'. Sometimes, the results from a gaming experience can be different from 'scientists' expectations as game participants bring 'reality into playing the game' as opposed to a 'stylized model'. During game sessions, it has been very common to hear stakeholders link game results to events in real life (e.g. …."oooh, could that be the reason, the river stopped flowing at point xxx"….."I see the reason why the rivers these days are brown these days compared to clean water we experienced during our primary school days, soil erosion from expanded cropland areas…". The involvement of catchment stakeholders right from game conceptualization to implementation, reverses the engagement to a 'bottom-up strategy'. The 'gamification' component is an important aspect, in our own opinion, any model can be 'gamified' to allow 'local stakeholders' to learn and understand the 'scientific models work (but scientists must be willing to 'unlearn' to 'learn'). It is this increased system understanding that can inform realistic catchment management plans.*

*Regarding the referee's comment that games can actually reify the 'wicked problem' on the basis of an outsider/expert view we argue is not the case with serious gaming. We argue that the outsider view is highly managed in the gaming sessions. **(See Ln 90)** The design of a serious game is an iterative process that evolves with the participative process whereby local stakeholders are actively involved in defining the 'wicked problem', and designing the questions, simulations, and outputs. During actual gameplay, the outsiders/experts are more 'facilitators' than the 'leads'. Hence the nature of engagement of participants is not as limited as in other conventional approaches e.g. workshops where scientists take the lead (see our responses to the community comment on this point).*

*We do not hold the view that there is  ONE "wicked problem" in the case study area. We have actually defined what we mean by 'wicked problems' **(see Ln 47-51)**. Just like any other modelling logic, the game tries to simplify the representation of the socio-environmental dynamics. We have stated that the ENGAGE game mimics the dynamics observed during the dry seasons in the upper Ewaso Ng'iro catchment. The downstream-upstream reactions result after the processes both in the socio and physical dynamics (it is not just a lack of coordination between upstream and downstream communities, but an array of possibilities or outcomes that can be observed with serious games).*

*We agree with the referee's comment that the critique of IWRM approaches would be more powerful and convincing if it were to be done at the level of how are the terms of the debate/participation set and by whom and the politics that framed the entire research process. As mentioned earlier, the outsider has no exclusive power to dictate the process. We have revised the manuscript to make this important aspect of serious gaming. Right from the conceptualization of the game, the local stakeholders have been involved in defining the key actors, resources, dynamics, and interactions. And this is one of the ways that games differ from other conventional participatory approaches.*

*The pre-print shared by the referee* ter Horst et al,. (2023), *actually re-affirms the politics around conventional approaches and modelling where the 'outsider' e.g. a hydrological scientist/modeler is the one defining the model components depending on the area of interest. This has been the norm for years, and conventional models have been used to shape discussions on water resources management and governance. It is very common to have presentations of model results e.g. during a workshop. Most of these models are 'black-boxed' and most of the specifics are left to the modelers. The opposite is true in serious gaming, the whole process is 'open' and defined in collaboration with stakeholders (scientists and non-scientists) at all stages, conceptualization, refining, and implementation. In other words, you cannot present or 'force' a game, that participants have no clue about or are not interested in (the game must mimic the real socio-environmental context of the targeted stakeholders).*

*We however recognize advanced game versions that go beyond a physical board game or use of cards into developing an Agent Based Model. ABMs can be helpful in understanding the 'system dynamics' through computer simulations, however, in situations where the interest is to explore ways of peaceful coexistence between worrying communities due to hydrological dynamics (i.e. strengthening engagements), the option of using 'real agents'/real participants in a 'safe environment' remains the better option since stakeholders can express their interests, concerns, test scenarios and explore possible way forwards. Therefore, investigating the potential of serious games in the context of strengthening stakeholders remains useful, which can be seen as advances in the field of water resources management and governance.*

**Revisions made in the manuscript:**

*We have revised the manuscript to make the case study context clearer. The following has text been included in the revised version (see Ln 224-226):*

*…..Violent conflicts exist between different water users at different levels: upstream versus downstream water users; competing irrigators; agro-pastoralists versus pastoralists; users versus authorities, environmentalists etc (Aarts, 2012; Ehrensperger & Kiteme, 2005)……*

*In addition, the following text has also been included in the revised manuscript (see Ln 244-246):*

*…… Given the context above and to explore possible alternatives to WRUA management styles (especially increasing stakeholder engagement), developing and testing an alternative participatory approach such as a serious gaming is timely……*

*We have revised the text to expound our argument and also reflect more on serious games short comings, including a reflection on the politics around conventional modelling and gaming, and expound on 'black-boxed' and 'open' approaches (see additional text in the revised manuscript Ln 93-127).*

**(2)** **MAJOR COMMENT 2: They could also engage a bit more with the risks/limitations of games that have been discussed in the literature**

Line 93 and following. The way to depicts serious games and what they can help achieve is overwhelmingly positive. I think this reflects rather well the existing literature on serious games, most of which is actually written by people involved in their design and implementation – which may make them a bit less likely to adopt a critical stance towards these tools. Yet, I think the authors could engage with some of the limits of games that have been discussed in the literature, notably by members of the ComMod collective they are likely to know. There is some work in relation to conflicts and existing relations of domination (Barnaud & Van Paassen, 2013; Bécu et al., 2008; Mathevet et al., 2014), on the roles and postures of facilitators and modelers (Barreteau et al., 2003; Jonsson et al., 2007), or the risk of manipulation (Barnaud et al., 2008; Halbe et al., 2018). A recent overview recognizes ''potential biases'' (Barreteau et al., 2021). See also (Jonsson et al., 2007) on whether games are "truly participatory". Also, the fact that games simplify complex problems (line 102) is presented as a positive thing (I suspect in relation to legibility of the said problem). This may be the case, but it also raises a lot of questions: who decides what is represented and what is not, what are the implications of those choices… In other words, what are the politics of the game itself?  These questions ought to be at least mentioned I think.

**Response:** *Thank you for the comment. This observation on positive look on serious games was also raised by the first reviewer. I think the context of the study area where there are complex water-related tensions and conflicts, may have led to the language used sounding positive. Especially given that the River Basin Organisations have been in existence for over 30 years (engaging stakeholders using the conventional engagement approaches) and the passion for exploring alternatives to stakeholder engagement may sound overwhelmingly positive. We have revised the manuscript to include some of the limitations reported in the literature.*

***Revisions made in the manuscript:***

The following text has been added in the revised version (see Ln 103-125):

*…… In their study, Flood et al. (2018) conducted a review of 43 serious gaming publications and identified the major shortcomings to effective game design and engagement as; one-off engagement (i.e. several game sessions are needed to enhance learning), capturing complexity without overwhelming the stakeholders, future planning (i.e. linking game results to plan an uncertain future). Serious games are also limited on the number of stakeholders who can be involved in a single game session, a constraint that raises the politics of who should attend the game(s) and why? (Wesselow and Stoll-Kleemann, 2018; Edmunds and Wollenberg, 2001). Studies have also reported that social differentiations and power asymmetries have greater influence on the outcomes of a participatory process (Barnaud and Van Paassen, 2013; Mathevet et al., 2014). Both the facilitators and the stakeholders have various degrees in which they can influence the participatory process (Jonsson et al., 2007). Serious gaming can also exacerbate the contests of power due to constraints of simplifying the complex real worlds, balancing the interests of the locals and the 'outsiders', and different perspectives of the present and future (Venot et al., 2022). A co-construction process also referred to as companion modelling approach where the designers and the participants collaborate to define the entire process is seen as a way to improve legitimacy of the participatory process and enhancing multi-stakeholder cooperation (Barnaud and Van Paassen, 2013; Barreteau et al., 2014; Basco-Carrera et al., 2018; Étienne, 2014). However, the companion modelling approach needs to be improved to clearly define the horizontal and vertical dialogues by involving all stakeholder at all levels (Barnaud et al., 2008). In general, the quality of participatory process depends on how biases and interests of all stallholders, including 'outsiders' are balanced (Biggs et al., 2021; Daniell et al., 2010). As aforementioned, the politics that shape conventional processes (e.g. the influence of the 'outsider') are dealt with in the gaming approach through an iterative process that evolves with participatory modelling (Marini et al., 2018a; Speelman, 2014a; Speelman et al., 2019b; Rodela et al., 2019b; Barreteau et al., 2014). Hence, this study can be viewed to have done something different from the conventional participatory approaches (such as workshops, where 'outsiders' dictate the process) by creating a different type of collaborative engagement and a 'safe environment' for stakeholders.……*

In relation to the question of who decides what is represented and what is not, we have revised the manuscript by adding the following (see Ln 93-102):

*……Compared to the conventional approaches and modelling, where the 'outsiders' (e.g. hydrological modellers and scientists) define the model components depending on the area of interest (Babel et al., 2019; Mayer et al., 2017), the 'outsiders' have no exclusive power to dictate the serious game components. While the conventional models are 'black-boxed' (Kouw, 2016; Melsen, 2022), the gaming process is 'open' and defined in collaboration with stakeholders (scientists and non-scientists) at all stages, from game conceptualization, game refining, to game implementation. This is one of major differences how serious gaming approach differs from other conventional participatory approaches such as workshops. There are different ways to increase engagement of participants during workshops, such as participatory mapping, experimentation with art-based visuals, etc, however, these cannot be viewed as collaborative modelling. Basco-Carrera et al. (2017), attempts to differentiate what can be considered as 'participatory modelling' and 'collaborative modelling'….*

**(3) **MAJOR COMMENT 3: Provide a bit more information on the methodology (including the link between the game and the computer model)**

I feel there is a need to clarify the methodology. Notably, in the first paragraph it is not clear to me what the "solution space" of the game (what are 'responses' to rules?). It is also not clear what the authors mean by "overall performance of the game results" (how is performance assessed, is it in relation to water availability/sharing? In relation to how active participation was…)? In relation to that, section 2.4.3 discusses a modeling of the 'game solution space' and mentions a 1000 runs but what are those runs? I assume the authors used both a physical game and a computer version of it and that they used the later a 1000 times generating random actions; this to situate the results of the specific game sessions in the "realm of possible". But I can only assume this: the relations between game and model need to be made explicit and the authors need to explain, why the use of the model was useful, what did it allow to do?

*Response: Thank you for the comment. In **Ln 366** we have described the Solution Space as …..an envelope within which ENGAGE games operate, by understanding the minimum and maximum values of the various game metrics…...(i.e. the space of minimum to maximum of game results). The 'overall performance' mentioned in **Ln 177**, refers to actual game results plotted within the Solution Space. In other words, the general trends observed in the different rounds of the game. The solution space was constructed within a system dynamics modelling environment by a total of 1000 runs. Hence, this should not be seen as an independent computer model version.*

***Revisions made in the manuscript:***

We have revised the manuscript to give a clarity on what we refer to as Solution space (see Ln 175-179):

*….The solution space of the board game elements was developed to determine the realm of possibilities of participant choices in the ENGAGE game. The possible ranges (the minimum and maximum limits of game results) were explored in the modeled solution space. The overall performance of the game was assessed by plotting the actual game results within the solution space….*

(4) **MAJOR COMMENT 4: The mechanics of the game in the core of the paper (fez readers are likely to read the supplementary material)**

A lot of key information on the mechanics of the game (including how the connections between actors and water flows are conceptualized) are included in the supplementary material. I think only few readers are likely to read the supplementary material and it might be a good idea if the authors extended the presentation of the game in the core of the paper slightly. I'm also not sure the game should be presented in a "box". Normal section numbering would do. In terms of presenting the game, I think it would be interesting to have a sub-section describing the board and the players, another one describing briefly the different action/decision each actor can make and how this impact water flows (at a conceptual level; not entering in the details of the calibration), and another one on the "typical" process followed during each session – this is already in large part in the box. I think the graph on the system dynamic modeling could support this presentation of the game, as opposed to be presented in relation to the "solution space" of the game. The supplementary file can then focus on the calibration aspect. Presenting the game as such would also allow having part of the discussion engaging with the key stage of game validation that is not really described at the moment (what hypothesis the authors made were validated, which one were not…? for instance deciding that water dries up from downstream when people abstract water from upstream is a strong choice. I would be curious to know if that triggered some discussion also because, materially, it involves taking a marble from one area of the board to put it in the other one, which is not really how water abstraction works in the real world.)

***Revisions made in the manuscript:***

*Thank you for the comment. We have revised the manuscript by repackaging the information in the Box 1 into the following sub-headings (see Ln 261-315):*

***Gameplay description and process***

*1.1. Description of Boardgame and players*
*1.2. Game mechanics*
*1.3. Key actions and key outcomes expected in the game*
*1.4. Potential impact on water resources availability*
*1.5. Possible reactions expected by actors and feedback*

*Regarding the comment on validation, this is described in **sub-section 2.3 (Game pre-testing and validation).** The type of validation described here is 'expert-based validation type' where the catchment stakeholders are involved to reinforce trust, ownership, and confirm the processes and game outcomes and assumptions made (i.e. legitimacy, credibility and salience of the ENGAGE game). The validation did not take the conventional route of testing hypothesis but collaboratively testing and refining the game to fit the context and interests of the targeted stakeholders.*

(5) **MAJOR COMMENT 5: Finally not only discuss the fact that the game seem to have enhanced communication among participants and their promises in general, but also issues related to the politics of the game themselves, that is, the extent to which the representation of the system was accepted/validated/discussed by participants and the scope for the game to change the broader rules of the game and conditions of operation of the WRUAs.**

The discussion focuses on what has happened during the game session and it is important. The game sessions are likely to have improved communication and cross learning among participants but to which extent did it actually provide ways to solve the structural issues faced by WRUA that are identified in line 188 (weak enforcement of policies/laws, water abstraction regulations, water metering requirements, protection of riparian corridors/forested areas, etc.)? Such structural issues are likely to relate to the institutional strength of the WUAs, how legitimate it is vis-à-vis other actors, the (human) and financial means it has… Even if it is not the objective of this paper to assess the scope of the game to change things in the real world, I think it is important for the

authors to engage in a discussion around the promises and potential of games in relation to these structural hurdles.

Most of the literature on games tend to focus on the design process and on what is happening during game sessions, and that's why most of the literature is positive: there is a lot of interesting things happening, some of which do have transformative potential. But in doing so, the game and the overall research process is also "removed" from the overall political-economic context in which it is implemented while this context is pivotal for the promises of the game to actually materialize or not. While describing the overall political-economy that influences water management in the case study area is beyond the scope of this paper, I think it is important the authors reflect a bit on it and how this might influence the future of the game outcomes – to avoid falling in the trap of highlighting promises that may never materialize.

***Revisions made in the manuscript:***

*We have revised the discussion to reflect more on these. See the added sentences below (Ln 564-570):*

*…..The WRUAs have been in existence for over 30 years employing conventional participatory approaches to engage catchment stakeholders, especially in minimizing human-water related tensions and conflicts. This study shows that the first two 'issue cycle' steps, agenda setting (acknowledging that there is a problem) and shared understanding of its causes and consequences, were readily addressed by the game in all three game sessions. As an alternative to existing conventional approaches, WRUAs can readily adopt the ENGAGE approach to engage catchment stakeholders in minimizing conflicts, promoting collectiveness and dialogues through active participation, increase knowledge on human-water interactions. The next steps on commitment to goals and means of implementation would depend on the way the game is part of a longer-term process of interactions….*

**I also have some specific comments:**

**Line 21 (Abstract):** What do the authors mean by "shared understanding", and understanding of what exactly? Could it be clarified?

**Response:** Thank you for the comment. The sentence have revised as follows:

….Lack of common understanding on human-water perspectives by catchment stakeholders increase the complexity of human-water issues at the river catchment scale…..

**Line 27 (abstract):** This sentence seems to point to the fact that the objective of the study was to assess the potential of the game to strengthen stakeholder engagement "next" (that is, after the game sessions were held). I do not think the paper demonstrates this. It demonstrates rather convincingly that there was active engagement during the game session but does not engage with whether or not this translated in strengthened engagement outside of the game session. I understand from the exchange with the community member who commented on the paper that this is not the point of this particular paper but then, maybe this sentence needs to be rephrased to indicate that what is being described in the dynamics of engagement during gaming session. Similarly, some clarity may be needed on line 125.

***Response:*** *Thank you for the comment. In this study, we did not design to have comparisons e.g. comparing engagements with and without gaming sessions. Rather, we focussed on engagement patterns observed during gameplay given the context described in the paper and in the above responses. In this paper we did not follow up the engagements outside the game environment, but three pilot game sessions provide important highlights that can help reflect on the potential of using serious gaming as an alternative approach to stakeholder engagement, given the context especially the use of conventional participatory approaches that have been in existence for over 30 years.*

We have revised the text to read as follows:

*….The purpose of this study was to explore the potential role of serious gaming in  subsequent steps of strengthening stakeholder engagement (agenda setting, shared understanding, commitment to collective action, and means of implementation) explored within a game environment toward addressing complex human-water-related challenges at the catchment scale…..*

**Line 47.** The term "wicked problems" was first coined by Horst Rittel and Melvin Webber in a 1973 paper, which would be good to mention.

**Response:** Thank you for the comment. This reference has been included in the revised version.

**Line 54 to 56:** It is not clear how the 5 interacting phases of public debate had a bearing on the research conducted. Why do the authors refer to this work, and how useful is it to understand the process of designing and implementing game sessions? This is clarified around line 95. Consider slight restructuring on when this is mentioned so as to avoid repetitions?

**Response:** Thank you for the comment. We have revised the sentence to give a clarity that there are five stages in engaging stakeholders in natural resource management as follows:

*…..Five interacting phases in the public debate on engaging stakeholders in natural resource management were identified as (a) agenda setting, (b) shared understanding, (c) commitment to goals, (d) means of implementation, and (e) re-evaluation based on monitoring….*

**Line 58 and following:** I would not call IWRM an "approach". In my view it consists rather in a policy and discursive model than an approach per se.

**Response:** Thank you for the comment. We have replaced "approach" with "process"

**Line 63:** can the authors briefly say what successes were actually achieved? In what terms was implementation successful and in which contexts?

**Response:** Thank you for the comment. We have revised the sentence to include the successes mentioned in literature as follows:

**Line 107:** improved efficiency of what?

**Response:** Thank you for the comment. This has been revised to improved efficiency of strategies

**Line 101:** what does "this" relate to?

**Response:** Thank you for the comment. I do not see 'this' in **Ln 101**, could you be referring to **Ln 112**? If this is the case, 'this' relates to 'studying communication patterns'…. The phrase has been revised accordingly.

**Line 112 and following:** I am not sure what those sentences are meant to bring to the paper. They look as quite general statements in relation to games (what others have said on games), but it is not clear how much of this is actually being used in this particular study. Maybe they do not belong here but should come earlier in the paragraph? I have a similar remark on communication: are the two sentences on that specific topic meant to discuss game "in general" or has this topic of communication during game played been a specific entry point of analysis of what has happened in the game session organized in this study?

*Response: Thank you for the comment. The sentences appear immediately after describing the importance of 'communication analysis' especially during engagement sessions. These sentences are important to reflect on in the introduction section because, during communication/engagements this could be in line with relational logic (value attached on how stakeholders relate to one another) or instrumental aspects (economic perspectives) or both. Furthermore, it would also be important to study communication patterns and emerging patterns by looking further into 'pressures' or 'external factors' influencing changes in strategies and decisions made by participants. For instance ' a participant revising strategies due to a government fine. Yes, the communication analyses focussed on what happened during game sessions.*

We have revised the sentences as follows (see Ln 144-157):

*…….Communication is one of the social parameters that enable the manifestation of a group strategy, improved efficiency of strategies, and better decision-making (references). In a serious gaming environment, communication during gameplay is a key factor influencing game outcomes (references). Hence, studying communication patterns during gameplay can help evaluate the stakeholders' engagement and interpret emergent game results. This contributes to the body of knowledge on using the serious gaming approach as an 'alternative tool' to addressing complex 'wicked' problems. Studying communication patterns can help study relational logic (value attached on how stakeholders relate to one another) or instrumental aspects (economic perspectives) or both (references). In addition, games can explore multiple levels of internalization of external impacts of individual decisions, based on rules, economic incentives, co-investment, peer pressure to reduce one's footprint, or genuine concerns for impacts on others (references). Games can pose a challenge to the*

*players who remain selfish as long as they only consider their direct interests, but emergent collective action can bring new solutions……*

**Line 132:** what do the authors mean by "sentiment" and how did they assess it. What do they mean by "active" participation and how was it assessed.

*Response: Thank you for the comment. Sentiments refers to 'verbalized statements' expressed during gameplay. As stated in **Ln 195**…. The game sessions were video recorded to allow post-game analysis of sentiments. Each of the extracted sentiment/statement was subjected to a scoring scale described in **Table 1**. Active participation refers to 'enhanced communication among participants'-which was assessed by analysing the type and direction of sentiments e.g. directed to other participants, to the facilitator or as a spontaneous reaction to game outcomes.*

We have added the word 'verbalized' to read as "verbalized sentiments"

**Line 145 and following:** why does the discussion focuses on "substractive dynamics" only and not on "constructive dynamics"?

**Response:** Thank you for the comment. It is not very clear what the referee means with only focussing on 'subtractive dynamics'. **Ln 145** is an extended description of 'counterfactual thinking' concept. For instance, conflicts and tensions would trigger thoughts about alternatives to these problems. If that is the case, then it is logical to assume that subtractive dynamics such as selfishness, tensions, conflicts would reduce with the build-up of constructive dynamics such as knowledge gain, collectiveness, cooperation, collaboration etc. We have not only focussed on subtractive dynamics, if you look at **figures 7** and **8**, we describe and discuss both subtractive and constructive dynamics.

**Line 160 and following:** how many WRUA are there in the sub basin/case study area? One for each sub-river?

**Response:** Thank you for the comment. There are seven WRUAS in the selected study area. Yes, one WRUA represent one sub-river area.

**Line 177 and following:** This points to my first major comment. How was this suite of problems identified and by whom? And more specifically, have the people who participated to the game sessions been involved in the definition of these problems. This needs to be clarified as it is largely from this that one can assess to which extent the process followed is fundamentally different from classic "participatory activities" implemented under IWRM processes.

**Response:** Thank you for the comment. These comment has been addressed while responding to major comment no. 1. Yes, the catchment stakeholders were involved in definition of game components, refining of rules and processes that would increase interest in playing the ENGAGE game. However, it is important to point out that the actual participants involved in the three pilot game sessions were different from those involved in conceptualization of the game. As the referee mentions in this comment, this is one of the aspects why serious gaming approach differs from other conventional workshops where the 'scientists' seem to take full control. In gaming concept, the 'scientists' play the role of 'facilitators' (from game conceptualization to actual implementation).

**References**

Di Baldassarre, G., N. Wanders, A. AghaKouchak, L. Kuil, S. Rangecrof, T.I.E. Veldkamp, M. Garcia, P.R. van Oel, K. Brein and A.F. Van Loon, 2018. Water shortages worsened by reservoir effects. Nature Sustainability 1, pp.617–622; https://doi.org/10.1038/s41893-018-0159-0

Waalewijn, P. (2002). "Squeezing the cow. A study on the perceptions and strategies of stakeholders concerning river basin management in the lower Komati River, South Africa," MSc thesis, Wageningen University, Wageningen.

Asmal, K.: Dams and development: a new framework for decision-making. The report of the World Commission on dams, Earthscan Publications Ltd, London, 2000.

Di Baldassarre, G., Wanders, N., AghaKouchak, A., Kuil, L., Rangecroft, S., Veldkamp, T. I. E., Garcia, M., van Oel, P. R., Breinl, K., and Van Loon, A. F.: Water shortages worsened by reservoir effects, Nat Sustain, 1, 617–622, https://doi.org/10.1038/s41893-018-0159-0, 2018.

Basco-Carrera, L., Warren, A., van Beek, E., Jonoski, A., and Giardino, A.: Collaborative modelling or participatory modelling? A framework for water resources management, Environmental Modelling & Software, 91, 95–110, https://doi.org/10.1016/j.envsoft.2017.01.014, 2017.

Irshad, M., Inoue, M., Ashraf, M., and Al-Busaidi, A.: The Management Options of Water for the Development of Agriculture in Dry Areas, Journal of Applied Sciences, 7, 1551–1557, https://doi.org/10.3923/jas.2007.1551.1557, 2007.

Kim, D., Stewart Carter, A. L., and Misser, S. A.: AA1000 Stakeholder Engagement Standard, Accountability, 1–40 pp., 2018.

Kuil, L., Evans, T., McCord, P. F., Salinas, J. L., and Blöschl, G.: Exploring the Influence of Smallholders' Perceptions Regarding Water Availability on Crop Choice and Water Allocation Through Socio-Hydrological Modeling, Water Resour Res, 54, 2580–2604, https://doi.org/10.1002/2017WR021420, 2018.

Sivapalan, M., Savenije, H. H. G., and Blöschl, G.: Socio-hydrology: A new science of people and water, Hydrol Process, 26, 1270–1276, https://doi.org/10.1002/hyp.8426, 2012.

Common Minimum Standards for Multi-Stakeholder Engagement in the UN Development Assistance Framework: https://unsdg.un.org/resources/common-minimum-standards-multi-stakeholder-engagement-undaf, last access: 6 December 2022.

Velasco-Muñoz, Aznar-Sánchez, Batlles-delaFuente, and Fidelibus: Rainwater Harvesting for Agricultural Irrigation: An Analysis of Global Research, Water (Basel), 11, 1320, https://doi.org/10.3390/w11071320, 2019.